# Monitoring peripheral nerve degeneration in ALS by label-free stimulated Raman scattering imaging

Feng Tian[1,2,3,*], Wenlong Yang[4,*], Daniel A. Mordes[1,2,3,5], Jin-Yuan Wang[1,2,3], Johnny S. Salameh[6], Joanie Mok[1,2,3], Jeannie Chew[7], Aarti Sharma[8], Ester Leno-Duran[1,2], Satomi Suzuki-Uematsu[1,2,3,†], Naoki Suzuki[1,2,3,†], Steve S. Han[1,2,3,†], Fa-Ke Lu[4,9], Minbiao Ji[4,†], Rosanna Zhang[1,2,3], Yue Liu[1,2,3,†], Jack Strominger[1,2], Neil A. Shneider[8], Leonard Petrucelli[7], X. Sunney Xie[4] & Kevin Eggan[1,2,3]

The study of amyotrophic lateral sclerosis (ALS) and potential interventions would be facilitated if motor axon degeneration could be more readily visualized. Here we demonstrate that stimulated Raman scattering (SRS) microscopy could be used to sensitively monitor peripheral nerve degeneration in ALS mouse models and ALS autopsy materials. Three-dimensional imaging of pre-symptomatic *SOD1* mouse models and data processing by a correlation-based algorithm revealed that significant degeneration of peripheral nerves could be detected coincidentally with the earliest detectable signs of muscle denervation and preceded physiologically measurable motor function decline. We also found that peripheral degeneration was an early event in *FUS* as well as *C9ORF72* repeat expansion models of ALS, and that serial imaging allowed long-term observation of disease progression and drug effects in living animals. Our study demonstrates that SRS imaging is a sensitive and quantitative means of measuring disease progression, greatly facilitating future studies of disease mechanisms and candidate therapeutics.

[1] Department of Stem Cell and Regenerative Biology, Harvard Stem Cell Institute, Harvard University, Cambridge, Massachusetts 02138, USA. [2] Department of Molecular and Cellular Biology, Harvard Stem Cell Institute, Harvard University, Cambridge, Massachusetts 02138, USA. [3] Stanley Center for Psychiatric Research, Broad Institute of MIT and Harvard, Cambridge, Massachusetts 02142, USA. [4] Department of Chemistry and Chemical Biology, Harvard University, Cambridge, Massachusetts 02138, USA. [5] Department of Pathology, Massachusetts General Hospital, Boston, Massachusetts 02114, USA. [6] Department of Neurology, University of Massachusetts Memorial Medical Center, Worcester, Massachusetts 01655, USA. [7] Department of Neuroscience, Mayo Clinic, 4500 San Pablo Road, Jacksonville, Florida 32224, USA. [8] Department of Neurology, Columbia University Medical Center, New York, New York 10032, USA. [9] Department of Neurosurgery, Brigham and Women's Hospital, Harvard Medical School, Boston, Massachusetts 02215, USA. * These authors contributed equally to this work. † Present addresses: Department of Transplantation, Reconstruction and Endoscopic Surgery, Division of Advanced Surgical Science and Technology, Tohuku University, Sendai 980-8572, Japan (S.S.-U.); Department of Neurology, Tohoku University School of Medicine, Sendai 980-8574, Japan (N.S.); Department of Neurology, Massachusetts General Hospital, Boston, Massachusetts 02114, USA (S.S.H.); Department of Physics, Fudan University, Shanghai 200433, China (M.J.); School of Life Sciences, Tsinghua University, Beijing 100084, China (Y.L.). Correspondence and requests for materials should be addressed to X.S.X. (email: xie@chemistry.harvard.edu) or to K.E. (email:eggan@mcb.harvard.edu).

Amyotrophic lateral sclerosis (ALS) patients suffer from spreading paralysis and terminal decline often within only 3 years of diagnosis[1,2]. Mouse strains carrying human disease-related mutations have emerged as important models for understanding the molecular and cellular events that underlie ALS[1,3] and phenocopy many processes observed in ALS patients including protein aggregation and motor neuron degeneration[3–6].

Improved methods for monitoring motor neuron degeneration are much needed, in part due to the high-degree of variability of disease progression in rodent models[2,7] and ALS patients[7–10]. End point analyses currently used to monitor disease progression in mouse models are also either laborious histological measurements or behavioural assays that can be impacted by operator-specific variance. Unbiased imaging methods could provide more precise measures of disease onset and progression, as well as reduce the length and ambiguity currently inherent to trials of candidate interventions in these models.

Stimulated Raman scattering (SRS) microscopy is an emerging chemical imaging technique that can map the distribution of molecules including lipids, proteins and nucleic acids in living cells and tissues based on their intrinsic molecular vibration[11–17]. SRS and its precursor coherent anti-Stokes Raman scattering have been demonstrated as a powerful tool for myelin imaging[18–26]. Although there is growing interest in the importance of demyelination in ALS, systematic studies of peripheral nerve degeneration have not yet been carried out with SRS[27].

Here we describe the use of SRS imaging to visualize peripheral degeneration in several mouse models of ALS and human postmortem tissue. We found that peripheral nerve degeneration as monitored by SRS imaging was one of the earliest detectable pathological events in ALS mouse models and that disease progression could be followed reliably over time in living animals through serial imaging. We also demonstrate that SRS imaging could be employed to evaluate candidate therapeutics, confirming that the compound minocycline significantly slows peripheral nerve degeneration in the SOD1G93A mouse. To demonstrate the potential clinical utility of our approach, we showed that motor nerve degeneration can similarly be monitored in postmortem tissue from ALS patients.

## Results

**SRS imaging of the sciatic nerve.** In SRS imaging, two focused beams of synchronized picosecond pulsed lasers (pump and Stokes) are overlapped in space and time on the sample to produce SRS signal[11–13]. The energy difference of the photons of the two beams can be tuned to match the vibrational frequencies of molecules to be imaged. As peripheral nerves are characterized by a myelin sheath with a high lipid content, we optimized parameters to allow imaging of lipids in fixed, isolated sciatic nerves (Methods section, Fig. 1, Supplementary Fig. 1)[11,19–26]. Collection of pump light, after the lasers' SRS interaction with the sample in focus, was achieved by the use of a high numerical aperture condenser and band-pass filter before detection with a photodiode (Methods section and Supplementary Fig. 1). After signal processing, the myelin sheath and nodes of Ranvier were clearly visualized in the nerve of wild-type (WT) animals (Fig. 1d,e) at an improved resolution and ease of execution relative to standard histological methods (Supplementary Fig. 2).

To explore whether SRS imaging might be useful for studying peripheral nerve degeneration in ALS mouse models, we imaged the sciatic nerve from the ventral root to the tibial nerve branch in 16-week-old SOD1G93A mice exhibiting end-stage paralysis (Fig. 1b). We found myelin sheath discontinuity and the appearance of oval structures in these animals that seemed to be derived from degenerating nerve fibers (Fig. 1e). These 'ovoid' structures were ubiquitous throughout the regions imaged

and were similar to those observed in models of nerve injury and Wallerian degeneration by histology and electron microscopy (Fig. 1b,e and Supplementary Figs 3,4a)[28–31]. The three-dimensional (3D) reconstruction of z-stack SRS images demonstrated that these ovoids were primarily irregular ellipsoids (Fig. 1f–j).

To determine how early degeneration in the SOD1G93A mouse model could be detected, we performed SRS imaging on dissected sciatic nerves at 4, 8, 12 and 16 weeks in transgenic males and their WT littermates (Fig. 1a). This timecourse suggested early changes in the peripheral myelin of SOD1G93A mice, as early as 4 weeks, numerous structural changes by 8 weeks and continued degeneration as animals aged (Fig. 1d–g, Supplementary Figs 3 and 5). Similar changes were observed in a smaller cohort of 12-week-old trasngenic female mice (Supplementary Fig. 3b).

**Ovoid structures are primarily lipids.** To further understand the ovoid structures appearing in SOD1G93A nerves, we performed whole-mount immunohistochemistry with antibodies specific to neurofilament (an axonal protein), myelin basic protein (MBP), an antigen produced by Schwann cells, and CD45, which marks infiltrating macrophages on nerve injury[32]. We then attempted to visualize ovoid structures by SRS followed by imaging of immunofluorescence by confocal microscopy (Methods section). We found that a number of ovoids in the SRS channel were labelled by anti-MBP antibodies, while there was negligible overlap between SRS signal and neurofilament or CD45 staining (Fig. 2a, Supplementary Fig. 6).

To further characterize the chemical composition of ovoids, we conducted a detailed SRS spectral scan in both WT and SOD1G93A sciatic nerve. Myelin is mainly composed of cholesterol, lipid, protein and water[33–35]. The lipid components of myelin include galactocerebroside, sphingomyelin, phosphatidylcholine and phosphatidylethanolamine[35]. We used the least square method to dissociate the spectra of myelin and degenerated myelin to the seven chemicals mentioned above (Methods section). We found that ovoids had a chemical composition similar to myelin, but with a trend towards an increased abundance of lipids and a significant decrease in water and cholesterol (Fig. 2b). We, therefore, concluded ovoid structures were likely derived from myelinating cells and due to their high lipid content we termed them 'lipid ovoids'.

**Quantification of myelin degeneration.** We next established a computational method for quantifying lipid ovoids as a means of monitoring peripheral nerve degeneration in longitudinal optical sections. We selected three distinct lipid ovoids with typical sizes and shapes to serve as templates (Supplementary Fig. 7a). These templates were then used to generate spatial correlation maps for each of the SRS images to be quantified. Areas showing strong correlation to the templates were identified as regions of interest (ROI; Supplementary Fig. 7b and Methods section). ROIs that were found to have an area within a pre-established range (size larger than 7.4, 14.8 and 13.0 $\mu m^2$, respectively) and that were sufficiently round (circularity thresholds for the three templates were 0.75, 0.80 and 0.55, respectively) were scored as lipid ovoids (Supplementary Fig. 7b).

Quantification was repeated for each of the three templates in the 20 optical slices in a given z-stack (Supplementary Fig. 7b). By summing all counts in a given z-stack, we could calculate the total number of ovoids present in a given location. Using this approach, we quantified myelin degeneration in SOD1G93A mice and WT littermates between 4 and 16 weeks of age. Consistent with the qualitative changes we observed by manual inspection, there was a trend for an increase in the number of lipid ovoids in

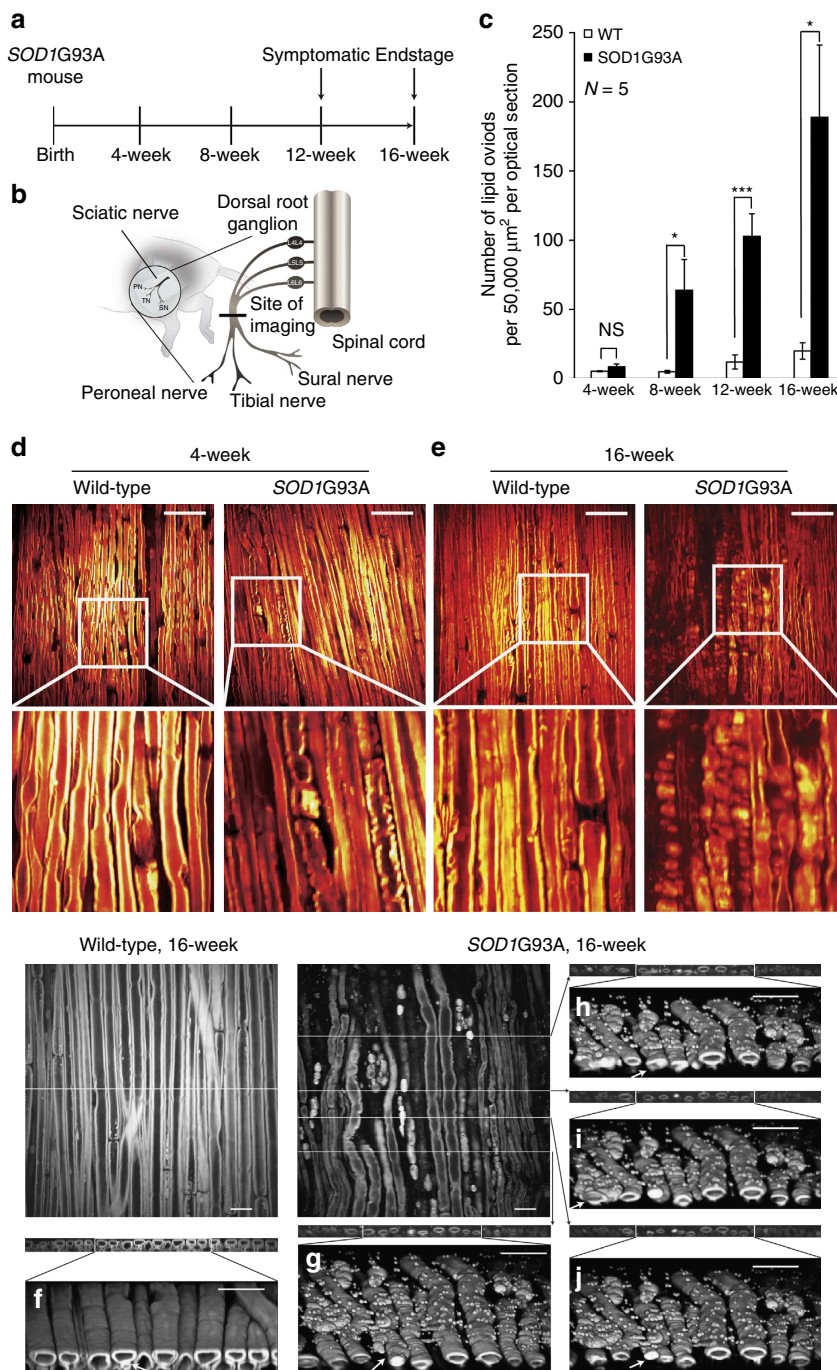

**Figure 1 | Age-point *ex vivo* SRS imaging of sciatic nerves identified early lipid ovoid deposition in *SOD1*G93A mouse ALS model. (a)** Experimental design of age-point *ex vivo* SRS imaging. **(b)** Sciatic nerve dissection from perfused *SOD1*G93A transgenic mice or WT littermates and the site of SRS imaging. **(c)** Lipid ovoid quantification result during disease progression time window of *SOD1*G93A mice. $*P < 0.05$, $***P < 0.001$, NS is not significant. $N = 5$ for each strain or each age point. Data are presented as mean ± s.e.m., and error bars show s.e.m. **(d,e)** *Ex vivo* sciatic nerve SRS images from *SOD1*G93A transgenic versus WT non-transgenic mice. Scale bar, 50 μm. **(f–j)** 3D reconstruction of lipid ovoids at different cross sections. Structures likely to be the wrapping vesicles can be visualized in the WT 3D reconstruction (the arrow in **f**). These ovoids had a variety of sizes, shapes and structures inside. The width of these ovoids was below 15 μm, which was just smaller than that of nearby myelin sheath. The length of ovoids varies dramatically. Larger lipid ovoids can be around 40 μm long (the arrow in **g**), while smaller ones were almost round in shape. Some of the lipid ovoids were empty inside (the arrow in **i**), some were solid (the arrow in **j**), while others were half empty (the arrow in **h**) seemingly with unidentified structures inside. Scale bar, 20 μm.

4-week-old *SOD1*G93A mice ($9 \pm 2$ ovoids per 50,000 μm$^2$ per optical section for *SOD1*G93A versus $5 \pm 0$ ovoids per 50,000 μm$^2$ per optical section for WT, $N = 5$ for each strain, $P = 0.13$; Fig. 1c) that became significant at 8 weeks of age ($64 \pm 22$ ovoids per 50,000 μm$^2$ per optical section for *SOD1*G93A versus $5 \pm 1$ ovoids per 50,000 μm$^2$ per optical section, $N = 5$ for each strain, $P = 0.008$; Fig. 1c). As suggested by manual inspection, the extent of myelin degeneration was greater and more significant in *SOD1*G93A mice at 12 weeks ($103 \pm 16$ ovoids per 50,000 μm$^2$ per optical section for *SOD1*G93A versus $12 \pm 5$ ovoids per

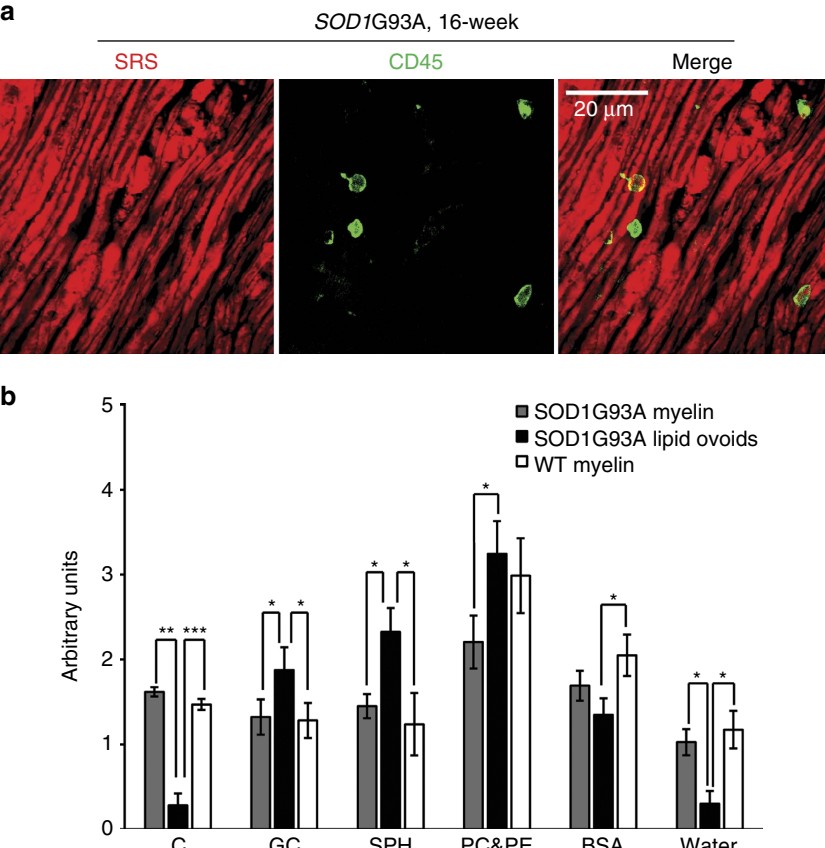

**Figure 2 | Characterization of lipid ovoids suggests that they were composed of myelin lipids. (a)** SRS lipid imaging and two-photon excited fluorescence imaging of CD45-Alexa 488 stained sciatic nerve from *SOD1*G93A 16-week end-stage mouse. Scale bar, 20 µm. **(b)** Chemical spectral analysis of lipid ovoids. *$P < 0.05$, **$P < 0.01$, ***$P < 0.001$. Analysed chemicals have been abbreviated as cholesterol (C), galactocerebroside (GC), sphingomyelin (SPH), phosphatidylcholine (PC), phosphatidylethanolamine (PE) and bovine serum albumin (BSA). Data are presented as mean ± s.e.m., and error bars show s.e.m.

50,000 µm$^2$ per optical section for WT, $N = 5$ for each strain, $P = 0.00015$; Fig. 1c) and 16 weeks ($189 \pm 52$ ovoids per 50,000 µm$^2$ per optical section for *SOD1*G93A versus $20 \pm 6$ ovoids per 50,000 µm$^2$ per optical section for WT, $N = 5$ for each strain, $P = 0.012$; Fig. 1c).

As a second means of quantification, we attempted to mimic a classical approach for monitoring peripheral myelin degeneration, the counting of intact nerve fibers in cross section[36]. To begin, we reconstructed cross sectional images of sciatic nerves from z-stack images (Fig. 1f, Supplementary Fig. 7c). Using SRS imaging, a band-pass Fourier filter and contrast enhancement to improve the reconstructed cross sectional images, we found that we could reconstruct nerve fibers present near the surface of the sciatic nerve (Fig. 1f, Supplementary Fig. 7c). A number of representative normal and blocked nerve fibers were selected as templates and we applied a method similar to that used for lipid ovoid quantification to count the number of blocked and intact nerve fibers. This quantification method could detect a difference between WT and *SOD1*G93A animals. However, significance was not reached until 16 weeks of age ($39.4 \pm 13.0\%$ for *SOD1*G93A versus $5.7 \pm 3.0\%$ for WT, $N = 5$ for each strain, $P = 0.0030$; Supplementary Fig. 8). Although these cross-sectional measurements were less sensitive than the longitudinal measurement of ovoids, they were consistent with historical counts, which detected alterations in myelinated fibers at a similar time point[36].

We next asked whether degeneration occurred with any spatial preference along *SOD1*G93A sciatic nerves. *SOD1*G93A mice at 8 postnatal weeks and their WT littermates were studied to capture early changes in myelin (Fig. 1c). All microscopic fields of the freshly perfused sciatic nerves were acquired in the lipid channel and integrated by an image-processing algorithm (Methods section). The resulting images provided a high-resolution global view of myelin sheaths where any detailed regions along the nerve could be individually magnified and examined. (Supplementary Fig. 7). Automated quantification was then employed to count the planar lipid ovoid numbers of eight assigned sub-sections from the proximal to the distal end of the nerve.

We found that sites of degeneration were distributed throughout the 8-week *SOD1*G93A sciatic nerve (Supplementary Fig. 9aiv–vi). In contrast, the WT sciatic nerve showed minimal signs of damage (Supplementary Fig. 9ai–aiii). Both the proximal and the distal regions of *SOD1*G93A sciatic nerve manifested a significant increase of lipid ovoid deposition over the WT sciatic nerve ($N = 4$, $P = 0.0084$ for proximal region, $P = 0.0018$ for distal region; Supplementary Fig. 9b). We did observe a trend for increase in lipid ovoids in distal regions of the nerve, although it was not significant ($N = 4$, $P = 0.20$; Supplementary Fig. 9b) and was mirrored by a similar trend for an increased number of distal ovoids in controls.

**SRS imaging in vivo.** We next explored the feasibility of serial imaging in ALS mouse models by attempting to image the sciatic nerve *in vivo* using a modified optical path (Supplementary Fig. 10a). As an initial end-point imaging study, mice at 8 weeks, 12 weeks and 16 weeks, respectively were anaesthetized and immobilized on the microscope stage (Fig. 3b and Methods

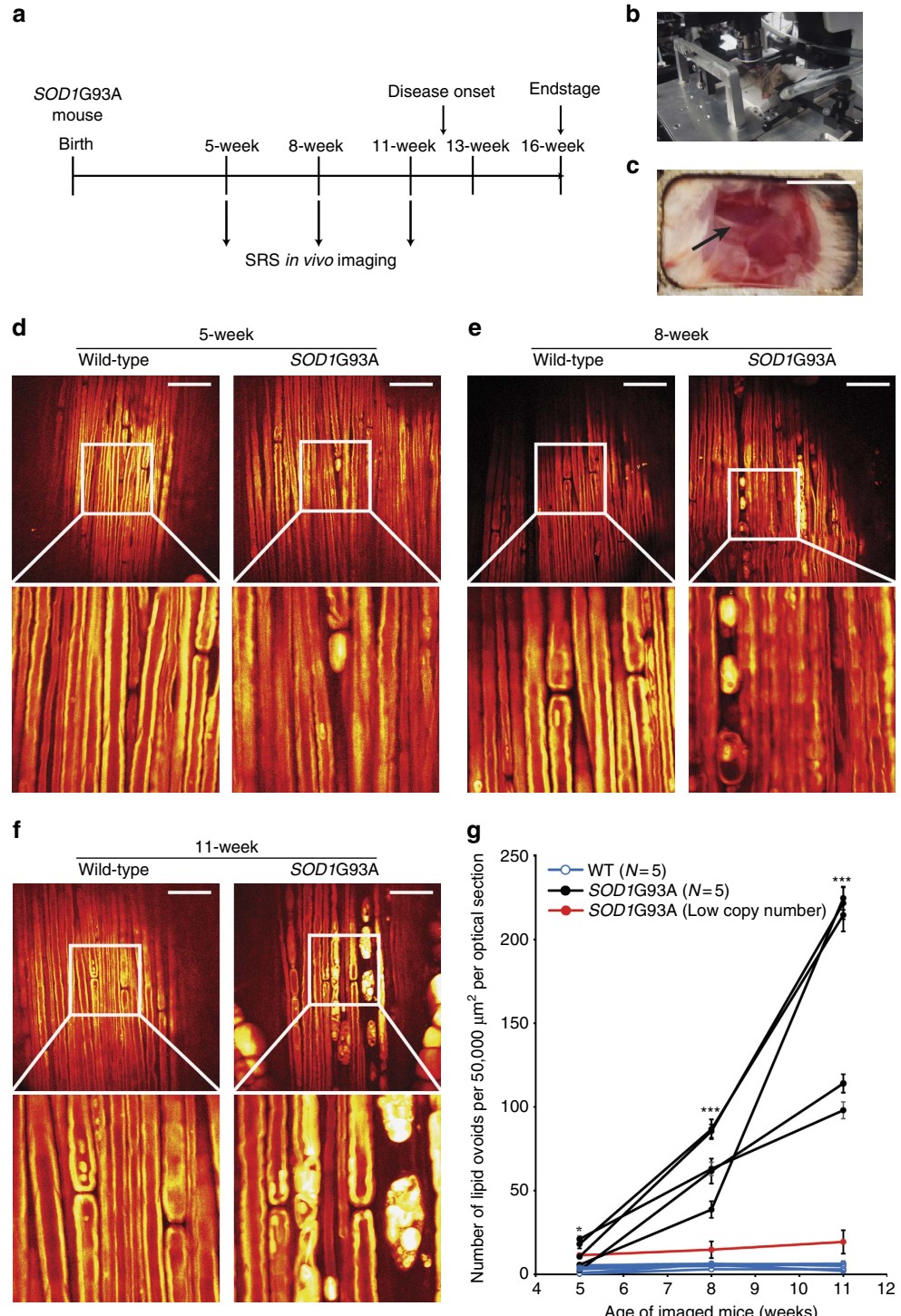

**Figure 3 | Progressive lipid ovoid deposition was visualized by long-term serial *in vivo* SRS imaging.** (**a**) Experimental design of long-term serial *in vivo* SRS imaging. (**b**) Stage alignment for *in vivo* SRS imaging of sciatic nerves. (**c**) Incision size of a representative imaged mouse. Scale bar, 1 cm. (**d–f**) *SOD1*G93A versus WT long-term serial *in vivo* SRS images at different ages during disease progression. Scale bar, 50 µm. (**g**) Quantification of lipid ovoids for long-term serial *in vivo* SRS imaging throughout disease progression. *$P < 0.05$, ***$P < 0.001$. $N = 5$ for each strain at each time point. For 5-week-old mice, $P = 0.043$. For 8-week-old mice, $P = 0.00012$. For 11-week-old mice, $P = 0.00031$. Data are presented as mean ± s.e.m., and error bars show s.e.m.

section). A small incision was made on the hind limb to expose the sciatic nerve and imaging was carried out avoiding muscle damage (Fig. 3c). To remove motion artifacts from respiring animal, we replaced any movement-distorted portion of images with the average of the previous and the subsequent optical slices (Supplementary Fig. 11). We again found a significant difference between *SOD1*G93A and WT mice at least as early as 8 weeks ($74 \pm 23$ ovoids per 50,000 µm$^2$ per optical section, $N = 5$ for *SOD1*G93A versus $4 \pm 1$ ovoids per 50,000 µm$^2$ per optical section, $N = 4$ for WT, $P = 0.031$; Supplementary Fig. 12).

Following our initial success with *in vivo* imaging, we attempted to monitor the rate of myelin degeneration in

individual animals by performing serial SRS imaging. We selected several time points for long-term survival imaging in attempt to capture the earliest onset of lipid ovoid deposition (Fig. 3a). We found that repeated imaging was well tolerated (Supplementary Fig. 10b, Supplementary Movie 1) and that imaging did not damage the nerve of control animals over the time investigated (Fig. 3d–g).

In contrast, the progression of nerve degeneration could be readily monitored in SOD1G93A mice (Fig. 3d–f). Lipid ovoid number was significantly increased in SOD1G93A mice as a group at 5 weeks postnatal ($12 \pm 4$ ovoids per $50,000 \, \mu m^2$ per optical section for SOD1G93A versus $3 \pm 1$ ovoids per $50,000 \, \mu m^2$ per optical section for WT, $N = 5$ for each strain, $P = 0.043$; Fig. 3d,g). By 8 weeks postnatal, each of the individual SOD1G93A animals exhibited a significantly increased number of lipid ovoids relative to the average of WT animals ($67 \pm 9$ ovoids per $50,000 \, \mu m^2$ per optical section for SOD1G93A versus $5 \pm 1$ ovoids per $50,000 \, \mu m^2$ per optical section for WT, $N = 5$ for each strain, $P = 0.00012$; Fig. 3e,g). This significant increase in myelin degeneration was even greater at 11 weeks of age ($175 \pm 28$ ovoids per $50,000 \, \mu m^2$ per optical section for SOD1G93A versus $4 \pm 1$ ovoids per $50,000 \, \mu m^2$ per optical section, $N = 5$ for each strain, $P = 0.00031$; Fig. 3g).

Interestingly, one SOD1G93A animal exhibited a much slower disease progression rate (Fig. 3g). This outlier was later confirmed by quantitative PCR to carry a lower copy number of the human SOD1G93A transgene[37] (Supplementary Fig. 13), which was consistent with previous observations that low copy-number SOD1G93A mice showed delayed and less-severe phenotypes[37].

**Ordering events in nerve degeneration**. To further understand the timing of peripheral myelin degeneration relative to other events occurring during disease progression in ALS mouse models, we performed both SRS imaging and electromyography (EMG) in 5-week-old animals[38]. Consistent with previous studies, quantitative measures of motor neuron physiology including compound motor action potential (CMAP), motor unit size and motor unit number estimation (MUNE) did not show significant differences between 5-week-old SOD1G93A and WT mice (Fig. 4a–c and Supplementary Table 1)[38]. In contrast, five of six SOD1G93A mice imaged by SRS exhibited an increased number of lipid ovoids relative to the average of their WT littermates. Furthermore, the overall comparison between the SOD1G93A and WT cohorts showed a significant increase in lipid ovoid number within ALS animals ($P = 0.034$; Fig. 4d).

By needle EMG, three out of six SOD1G93A mice within the same cohort displayed active denervation with fibrillation potentials and positive sharp waves, while all WT littermates were negative (Fig. 4e). These results confirmed that fibrillation potentials and positive sharp waves in needle EMG are one of the earliest detected minimally invasive biomarkers in the SOD1G93A mouse model[2,38]. Using SRS imaging, degenerative myelin morphology was observed in 4 out of 6 SOD1G93A mice, which again was not observed in any WT controls (Fig. 4f). Thus our results indicated that peripheral nerve degeneration as measured by SRS imaging reached significance at a time comparable to the initial observation of muscle denervation by EMG and weeks before the first detectable decline in motor nerve function. This early onset of nerve degeneration was confirmed in larger group of control and SOD1G93A animals at 6 weeks of age (Supplementary Fig. 14).

**Evaluation of minocycline in SOD1G93A mice**. The antibiotic minocycline has previously been reported to significantly extend the lifespan of SOD1G93A mice[39], though it later failed to show

efficacy in clinical trial[40]. We wondered whether SRS imaging was sufficiently quantitative and sensitive to be useful in revisiting the efficacy of this compound in a mouse model. As we had reproducibly observed peripheral nerve degeneration at 5 weeks of age by SRS, we initiated minocycline treatment at this age. SOD1G93A animals were randomly assigned to either minocycline ($N = 15$) or vehicle treatment ($N = 13$) groups, with daily administration via intraperitoneal injection (Methods section). In addition to control SRS imaging at 5 weeks, just before initial drug administration, we performed two later survival SRS imaging sessions on each animal in the study at 8 and 11 weeks of age (Fig. 5a).

Before the onset of drug administration (5 weeks), we found no difference between the overall number of lipid ovoids in transgenic animals randomized to the minocycline and vehicle treatment groups ($11 \pm 3$ ovoids per $50,000 \, \mu m^2$ per optical section for minocycline treated versus $10 \pm 2$ ovoids per $50,000 \, \mu m^2$ per optical section for vehicle treated, $P = 0.82$; Fig. 5c and Supplementary Fig. 15a), while both surpassed that of WT animals of the same age ($3 \pm 1$ ovoids per $50,000 \, \mu m^2$ per optical section). In contrast, following 3 weeks of minocycline treatment animals receiving minocycline showed a significantly lower burden of lipid ovoids compared with the vehicle controls ($30 \pm 9$ ovoids per $50,000 \, \mu m^2$ per optical section for minocycline versus $62 \pm 12$ ovoids per $50,000 \, \mu m^2$ per optical section for vehicle, $P = 0.036$; Fig. 5b,c), though both groups were elevated over WT control animals ($5 \pm 1$ ovoids per $50,000 \, \mu m^2$ per optical section). This effect of minocycline was sustained and became more significant at the 11-week imaging time point ($64 \pm 10$ ovoids per $50,000 \, \mu m^2$ per optical section for minocycline versus $96 \pm 7$ ovoids per $50,000 \, \mu m^2$ per optical section for vehicle, $P = 0.012$; Fig. 5c and Supplementary Fig. 15b), although the lipid ovoid deposition in these animals was not completely rescued as demonstrated by comparison with WT animals ($4 \pm 1$ ovoids per $50,000 \, \mu m^2$ per optical section for WT at 11 weeks of age; Fig. 3g).

To rule out the possibility that mice with a lower copy number of the SOD1G93A transgene might have been inadvertently randomized to the minocycline group, we determined the relative copy number of the SOD1G93A transgene in each of the study animals. Although we did observed variation in SOD1G93A copy number within the overall cohort, we found that this variation was equitably distributed between the minocycline and vehicle controls, with no significant difference in copy number between the groups (Supplementary Fig. 15c). We next attempted to relate transgene copy number, drug effects and the extent of peripheral nerve degeneration as measured by lipid ovoid accumulation (Fig. 5d). Before initial drug treatment at 5 weeks, we found there was a positive correlation between transgene copy number and lipid ovoid accumulation that was similar in the two separate groups of animals.

When SOD1 copy number was incorporated into analysis of animals at 8 weeks, the positive correlation between SOD1 copy number and ovoid number remained, however, the overall frequency of lipid ovoids was significantly reduced across animals of all copy numbers in the minocycline group (Fig. 5e). A similar pattern was observed at 11 weeks (Fig. 5f). Thus SRS imaging facilitated the finding that minocycline was able to modestly but significantly reduce the burden of peripheral nerve injury regardless of transgene copy number.

**Imaging SOD1G37R mice**. To determine whether the early degenerative phenotype we observed were conserved in additional ALS models, we imaged dissected nerves from SOD1G37R transgenic mice[4]. These mice exhibit a later disease onset and slower rate of phenotypic progression than observed in

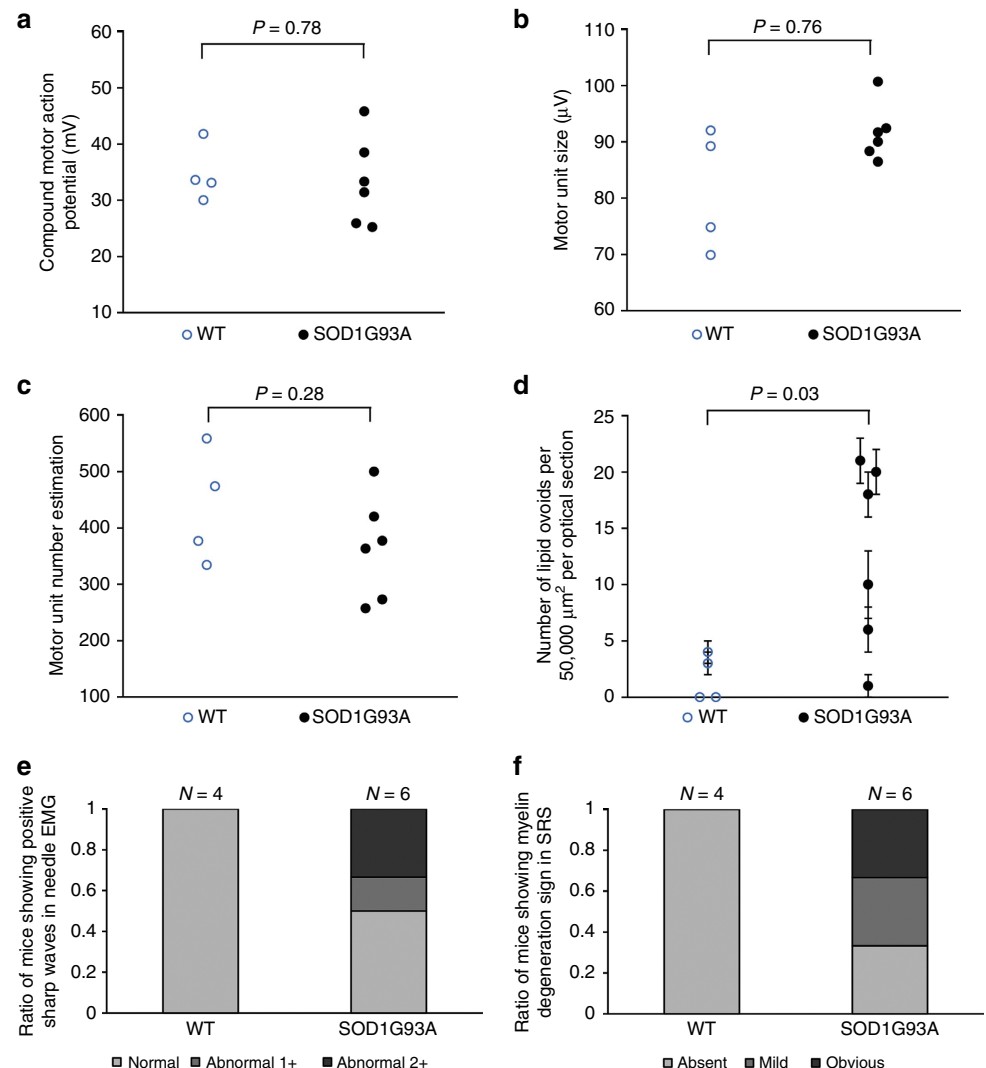

**Figure 4 | The sensitivity of SRS imaging was comparable to EMG for 5-week *SOD1*G93A versus WT mice.** (**a–c**) Quantitative parameters measured by EMG for 5 weeks *SOD1*G93A versus WT mice. (**d**) Quantification of lipid ovoids by SRS imaging for 5 weeks *SOD1*G93A versus WT mice. (**e**) Ratio of mice showing qualitative positive sharp waves by EMG for 5-week *SOD1*G93A versus WT mice. (**f**) Ratio of mice showing myelin degeneration signs by SRS imaging for 5 weeks *SOD1*G93A versus WT mice. $N = 4$ for *SOD1*G93A versus $N = 6$ for WT. Data are presented as mean ± s.e.m., and error bars show s.e.m.

*SOD1*G93A mice, making it a preferred model for some studies. Intriguingly, at 10 weeks of age, the earliest time point examined, we already observed a significant increase in degeneration relative to controls ($31 \pm 12$ ovoids per 50,000 μm² per optical section for *SOD1*G37R versus $4 \pm 1$ ovoids per 50,000 μm² per optical section for WT, $N = 4$ for each strain, $P = 0.043$; Supplementary Fig. 16d). This greatly preceded the earliest changes thus far reported by histological analysis (after 25 weeks)[4,27]. Further imaging at symptom onset (44 weeks) and end-stage (52 weeks) showed a further significant increase in ovoid numbers (Supplementary Fig. 16). We observed a number of lipid ovoids in end-stage *SOD1*G37R animals that was comparable to that we found in *SOD1*G93A mice ($239 \pm 17$ ovoids per 50,000 μm² per optical section for *SOD1*G37R $189 \pm 52$ ovoids per 50,000 μm² per optical section for *SOD1*G93A; Fig. 1c and Supplementary Fig. 16d).

**Non-*SOD1* mouse models of ALS.** We sought to employ SRS imaging in additional ALS models to expand its utility. The next model we examined was created by adeno-associated virus

(AAV)-mediated expression of the *C9ORF72* hexonucleotide repeat expansion throughout the central nervous system[5] (Fig. 6a). This *C9ORF72* model has been reported to exhibit neuronal RNA foci, dipeptide repeat protein inclusions, TDP-43 pathology, cortical and Purkinje cell loss as well as locomotive defects by 24 weeks[5]. By imaging *ex vivo* sciatic nerve samples of AAV-*C9ORF72* mice carrying pathogenic (G₄C₂)₆₆ repeats (66R) ($N = 12$) and their littermate controls carrying (G₄C₂)₂ repeats (2R) ($N = 13$) between 26 weeks and 28 weeks, we observed lipid ovoid deposition in the majority of 66R mice (Fig. 6b). Quantification demonstrated a significantly increased number of lipid ovoids in 66R animals ($29 \pm 6$ ovoids per 50,000 μm² per optical section for 66R versus $13 \pm 3$ ovoids per 50,000 μm² per optical section for 2R, $P = 0.024$; Fig. 6c).

We next examined a mutant *FUS* transgenic model in which either human WT or P525L mutant *FUS* is expressed from the mouse *MAPT* (tau) promoter (Fig. 6d). In this mouse model, It was been shown ~30% of motor neurons were lost and ~40% of tibialis anterior neuromuscular junctions were denervated at 52 weeks postnatal[41]. By examining sciatic nerves from h*FUS*P525L mice, we found significant lipid ovoid deposition was already

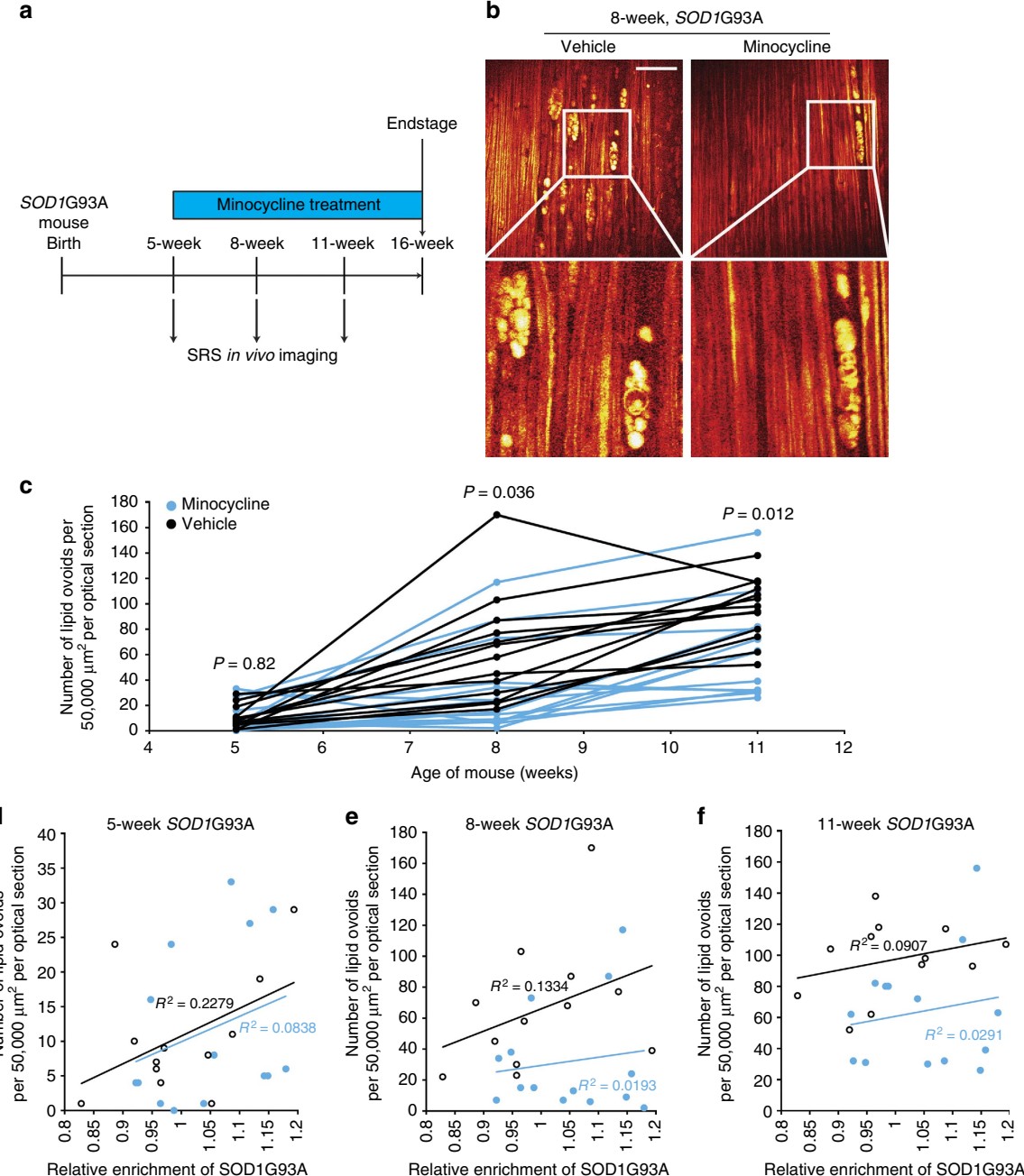

**Figure 5 | SRS imaging of *SOD1*G93A under minocycline treatment.** (**a**) Experimental design of long-term serial *in vivo* SRS imaging for *SOD1*G93A under minocycline treatment. (**b**) *SOD1*G93A versus WT long-term serial *in vivo* SRS images at 8 weeks postnatal. Scale bar, 50 μm. (**c**) Quantification of lipid ovoids for long-term serial *in vivo* SRS imaging throughout disease progression. (**d**–**f**) Correlation and regression analysis between *SOD1*G93A copy number and lipid ovoid deposition at 5 weeks (**d**), 8 weeks (**e**) and 11 weeks (**f**) of postnatal age.

present in these mice at 12 weeks of age ($43 \pm 6$ ovoids per $50,000 \, \mu m^2$ per optical section for h*FUS*P525L versus $15 \pm 2$ ovoids per $50,000 \, \mu m^2$ per optical section for h*FUS* WT, $P = 0.0008$; Fig. 6e,f). Overall, these results validate SRS imaging as a powerful resource to study peripheral nerve degeneration in emerging ALS mouse models.

**Imaging Sciatic nerve crush and EAE models.** We reasoned that if the lipid ovoids we observed in ALS models were resulting from injuring of the underlying neuron, then similar changes in myelin morphology might be observed distally to the site of nerve crush,

as a result of Wallerian degeneration. We therefore next tested the utility of SRS imaging in the context of a mouse model of axonal injury by sciatic nerve crush[22,23,26,42]. The sciatic nerve of one hind limb was crushed in WT and *SOD1*G93A animals then imaged distally of the injury at both 1 week and 3 weeks after initial injury (Fig. 7a).

We found WT animals exhibited significant peripheral nerve degeneration at 1 week ($662 \pm 287$ ovoids per $50,000 \, \mu m^2$ per optical section in crush versus $4 \pm 3$ ovoids per $50,000 \, \mu m^2$ per optical section in the uncrushed, $N = 4$ for each group, $P = 0.038$; Fig. 7b–d). At 3 weeks after injury, significant recovery was observed ($222 \pm 90$ ovoids per $50,000 \, \mu m^2$ per optical section for

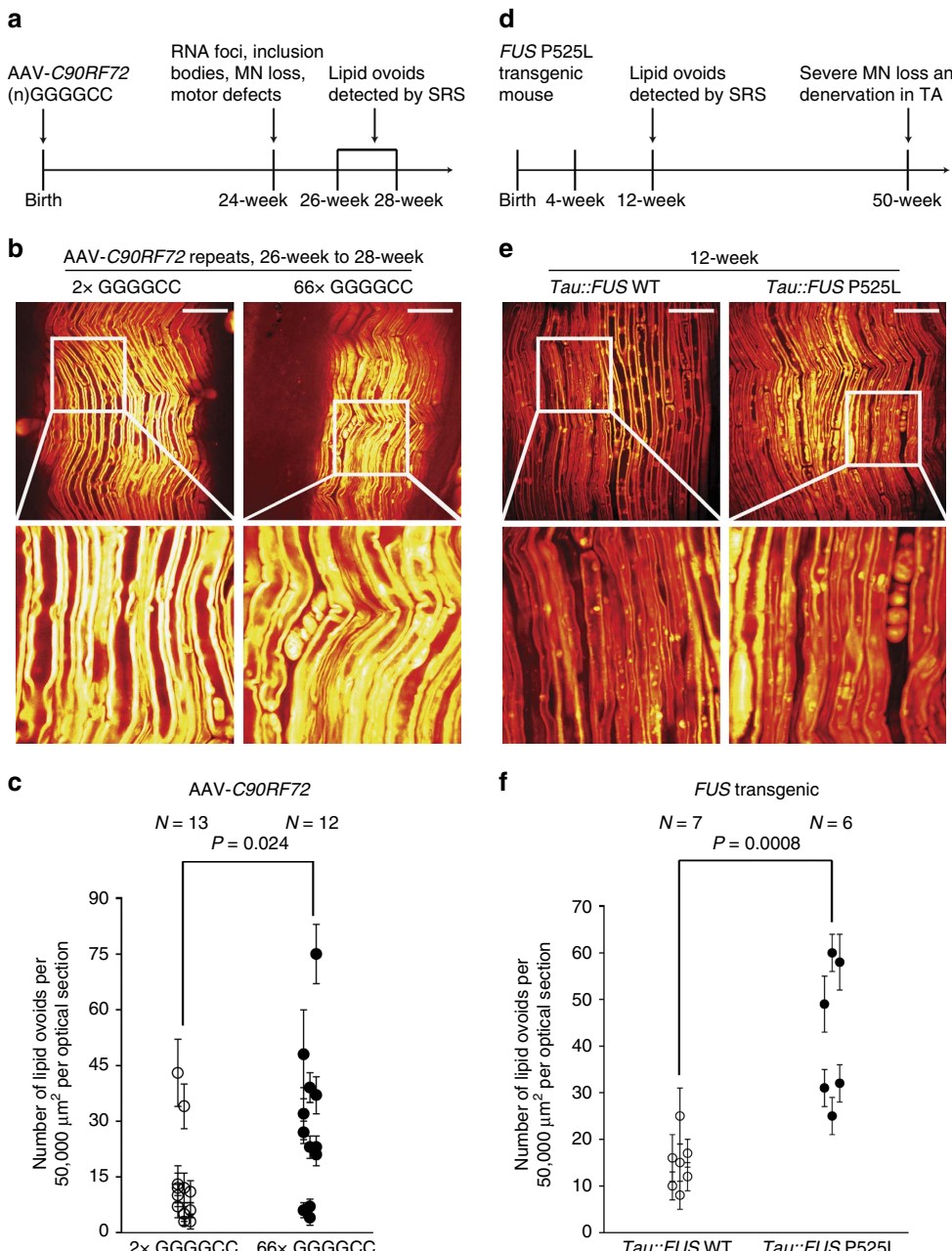

**Figure 6 | SRS imaging of non-*SOD1* mouse models demonstrated the general utility in studying ALS.** (**a**) Disease progression of AAV-C9ORF72 repeat expansion over-expression mouse model of ALS. (**b**) Representative SRS images of AAV-C9ORF72 mouse model of ALS showing healthy nerve fiber morphology for 2 repeats (2R) and lipid ovoid deposition (66R). Scale bar, 50 μm. (**c**) Lipid ovoid quantification result of AA-C9ORF72 mouse model of ALS. Data are presented as mean ± s.e.m., and error bars show s.e.m. (**d**) Disease-related phenotype progression in *FUS*P525L mouse ALS model. (**e**) *FUS*P525L versus *FUS* WT *ex vivo* SRS images at 12 weeks of age. Scale bar, 50 μm. (**f**) Quantification of lipid ovoids and statistical significance analysis for *FUS*P525L mouse model of ALS. Data are presented as mean ± s.e.m., and error bars show s.e.m.

3 weeks after crush, $N = 4$ for each group, $P = 0.038$ versus 1 week of recovery; Fig. 7b,d). We found that the morphology of degenerating fibers after nerve crush was very similar to that observed in *SOD1* mouse models, and when whole mount immunofluorescence was compared with SRS signal it again suggested that they were not likely to be infiltrating immune cells (Supplementary Fig. 17a).

We also examined the response to nerve crush in *SOD1*G93A mice and found that, with the exception of an elevated baseline of lipid ovoids relative to controls, these mice initially responded similarly to injury ($28 \pm 10$ ovoids per $50,000 \, \mu m^2$ per optical section for uncrushed versus $590 \pm 126$ ovoids per $50,000 \, \mu m^2$ per

optical section for crushed at 1 week, $N = 4$ for each group, $P = 0.038$; Supplementary Fig. 18). However, peripheral recovery in *SOD1*G93A mice was muted and did not reach significance ($590 \pm 126$ ovoids per $50,000 \, \mu m^2$ per optical section for 1 week after crush versus $295 \pm 135$ ovoids per $50,000 \, \mu m^2$ per optical section for 3 weeks after crush, $N = 4$ for each group, $P = 0.62$; Supplementary Fig. 18c).

We then compared the nature of peripheral nerve degeneration we observed in ALS and nerve crush models with that resulting from the damage of primary myelinating glial cells in the Autoimmune encephalomyelitis (EAE) model of multiple sclerosis (Fig. 7e)[43,44]. In late stage EAE animals, we clearly

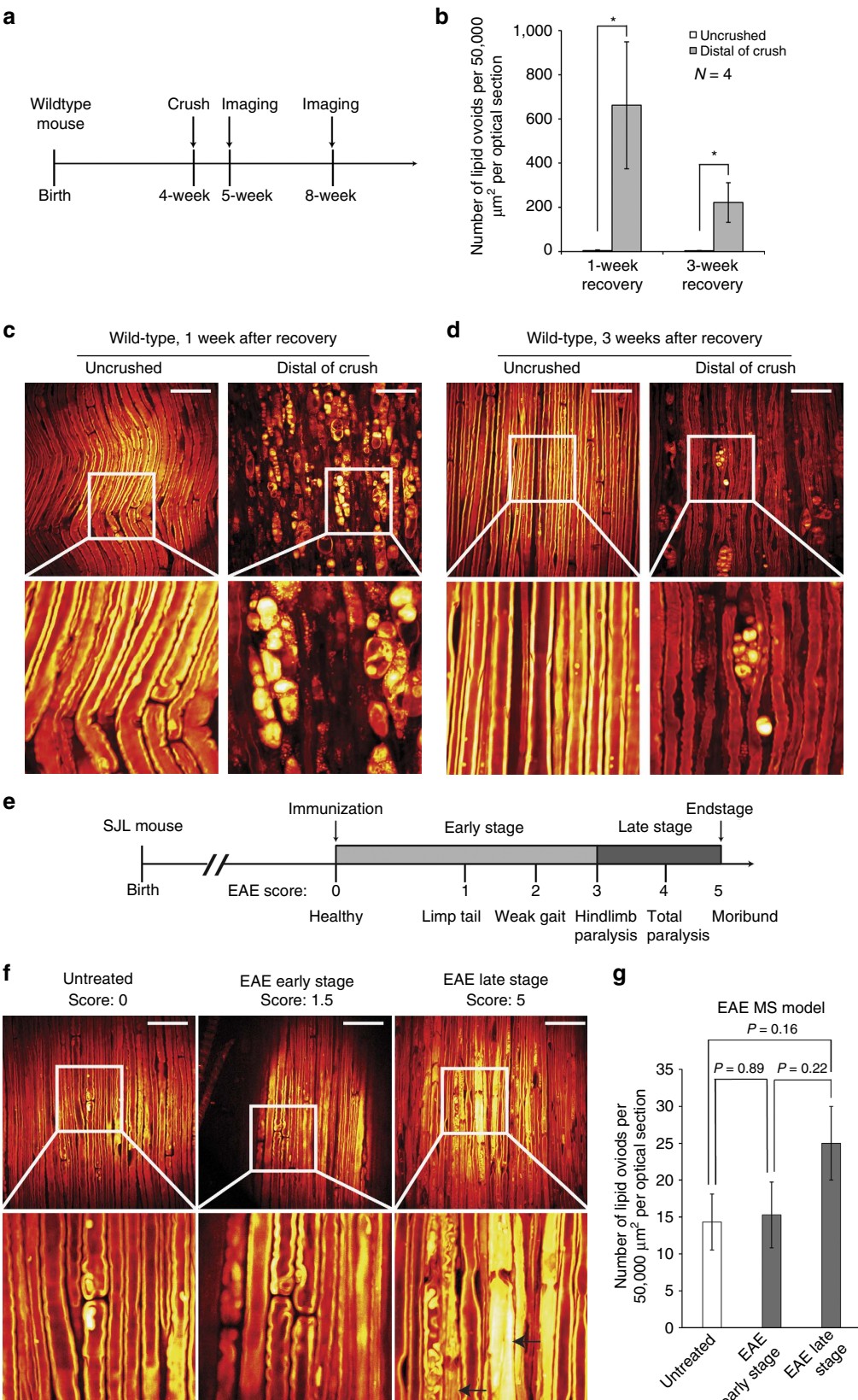

**Figure 7 | Lipid ovoid deposition could result from initial axon degeneration.** (**a**) Experimental design of SRS imaging for sciatic nerve crush model. (**b**) Lipid ovoid quantification result of sciatic nerve crush experiment. *$P < 0.05$. $N = 4$ for each strain or each age point. Data are presented as mean ± s.e.m., and error bars show s.e.m. (**c,d**) Sciatic nerve SRS images of WT mice after crush and recovery. Scale bar, 50 μm. (**e**) MS like phenotypes and disease progression of EAE model. (**f**) Sciatic nerve SRS images of EAE model. The left arrow shows demyelination morphology, and the right arrow shows remyelination morphology. Scale bar, 50 μm. (**g**) Lipid ovoid quantification result of EAE model SRS imaging.

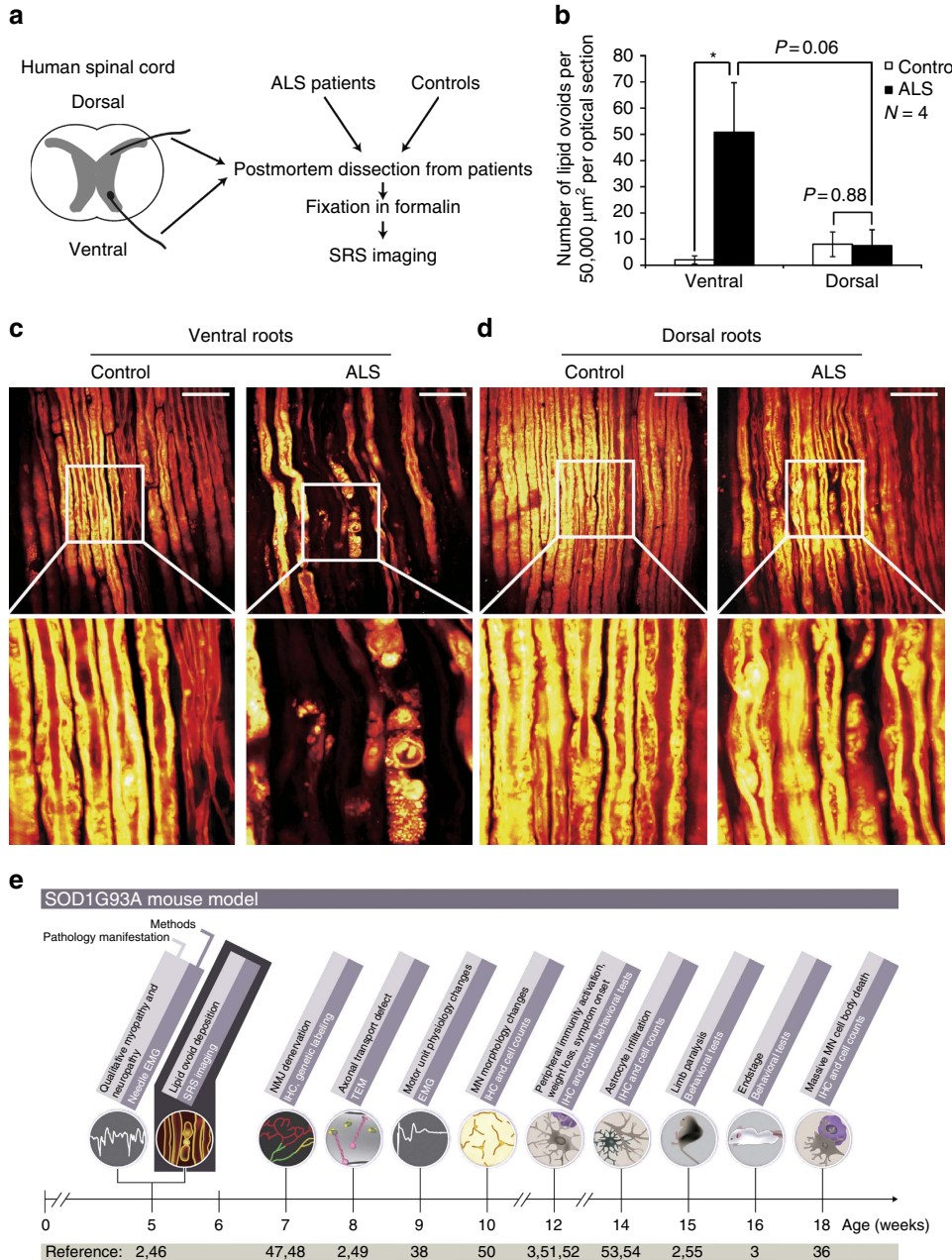

**Figure 8 | SRS imaging identified lipid ovoid deposition in human ALS patient samples.** (**a**) Experimental design of SRS imaging for human ALS patient samples. (**b**) Lipid ovoid quantification of ventral (motor fiber enriched) and dorsal (sensory fiber enriched) root samples from either human ALS patients or controls. *$P < 0.05$. $N = 4$ for each group. Data are presented as mean ± s.e.m., and error bars show s.e.m. (**c**) SRS imaging of ventral root nerve fibers from human ALS patients versus controls. Scale bar, 50 μm. (**d**) SRS imaging of dorsal root nerve fibers from human ALS patients versus controls. Scale bar, 50 μm. (**e**) Timeline of ALS associated pathological changes and diagnostic methods. EMG, electromyography; IHC, immunohistochemistry; MN, motor neuron; NMJ, neural muscular junction; SRS, stimulated Raman scattering; TEM, transmission electron microscope.

identified several sites where the morphology of myelin was altered (Fig. 7f). Following cross sectional reconstruction of these regions from optical slices, an apparent 'onion bulb' morphology could be visualized (Supplementary Fig. 19a). Our analysis indicated that relative to controls EAE nerves displayed an increased proportion of under-myelinated as well as over-myelinated nerve fibers (Supplementary Fig. 19b,c).

In contrast, we observed only a modest increase in lipid ovoid number even in moribund EAE animals ($14 \pm 4$ ovoids per 50,000 μm² per optical section for untreated versus $25 \pm 5$ ovoids per 50,000 μm² per optical section for EAE late stage, $N = 4$ for each group, $P = 0.22$; Fig. 7g and Supplementary Fig. 17). This

change was not comparable with the dramatic increase of lipid ovoids between *SOD1* end-stage and non-carrier animals. Our study demonstrated that ALS and MS models showed distinct peripheral myelin phenotypes.

**SRS imaging of human nerve samples.** We finally sought to validate the utility of SRS imaging in ALS patient samples[1]. To this end, we imaged post-mortem lumbar nerve samples from spinal cord ventral (motor) and dorsal (sensory) roots of sporadic ALS patients and controls (Fig. 8a and Methods section).

The quantification of z-stack SRS images demonstrated there had been substantial degeneration within the motor nerve fibers

from ALS patients ($51 \pm 19$ ovoids per $50,000\,\mu m^2$ per optical section for ALS ventral versus $2 \pm 2$ ovoids per $50,000\,\mu m^2$ optical section for control ventral, $N = 4$ for each group, $P = 0.040$; Fig. 8b,c). In contrast, no significant difference was observed between dorsal roots from ALS patients and those from controls ($7 \pm 6$ ovoids per $50,000\,\mu m^2$ optical section for ALS dorsal versus $8 \pm 5$ ovoids per $50,000\,\mu m^2$ optical section for control dorsal, $N = 4$ for each group, $P = 0.88$; Fig. 8b,d).

## Discussion

Here we show that SRS imaging can used to visualize peripheral nerve degeneration in ALS mouse models and post mortem materials. The sensitivity of SRS imaging was comparable to EMG for detecting early signs of neuronal degeneration, but also provided a richness of structural information about the peripheral nerve that EMG does not[45]. Strikingly, SRS imaging identified lipid ovoids in some SOD1G93A mice that showed no active denervation by EMG, suggesting that SRS alone or in combination with EMG may allow for the earliest *in vivo* detection of peripheral nervous degeneration in mouse models yet reported (Fig. 8e)[2,3,36,38,46–55].

In a study of minocycline, we found that SRS imaging enabled detection of the slowing of peripheral nerve degeneration induced by this compound. In the future, SRS could be used to potentially shorten the time frame for the evaluation of candidate ALS therapeutics as well as provide a more quantitative view of drug effects.

Our SRS imaging studies suggest that lipid ovoids accumulating in the peripheral nerves of pre-symptomatic ALS animals were most likely lipids derived from degenerating myelinating cells. We also visualized and quantified the peripheral demyelinating/remyelinating processes in the EAE model of MS, which were substantially distinct from those observed in ALS models[56]. The EAE model did not display any significant lipid ovoid deposition in peripheral nerves. Instead, we found that the pattern of degeneration in ALS mouse models and patient samples was more similar to that which occurs during Wallerian degeneration following axon injury[31,57]. Notably, it has been shown that introducing the Wallerian degeneration slow (Wld(s)) mutation into the SOD1G93A background slowed disease progression[58]. Similarly, forced expression of ATF3 (activating transcription factor 3), which is normally induced after sciatic nerve injury, also delayed muscle atrophy and extend the lifespan of SOD1G93A mice[57]. More intriguingly but less well substantiated, SARM1, whose activation has been observed after Wallerian degeneration, falls within a region of linkage disequilibrium associated with ALS[59]. The imaging studies reported here would seem to support the notion that signalling pathways involved in Wallerian degeneration warrant further attention in ALS.

With the development of fiber laser SRS and SRS based histological methods for studying brain tumours, the possibility of moving SRS microscope into a surgery room or clinic for rapid label-free histology is being explored[60–63]. Our studies have shown that SRS imaging might also eventually have utility for pathologists in rapid analysis of peripheral postmortem samples from ALS patients. However, given the current state of SRS technologies the greatest immediate utility of SRS imaging will likely be in the study of pre-clinical animal models.

Our natural history study of peripheral nerve injury in a variety of ALS models by SRS imaging further indicates that substantial nerve degeneration has occurred before the disease-related changes in the outward behavioural measures that are often used to determine disease onset and the commencement of drug administration studies in SOD1 models. As a result, the

therapeutic window remaining at the later time when many studies normally begin may be small, which could explain why detection of drug effects in these models has often been difficult. Incorporating SRS imaging as a means of determining the earliest onset of neural degeneration and then proceeding with initiation of treatment may allow for improved sensitivity in detecting the effects of experimental interventions.

## Methods

**Mouse models of ALS.** The mouse ALS models used in this study were the strain carrying human ALS-associated transgene SOD1G93A (B6SJL-Tg(SOD1*G93A) 1Gur/J, JAX Laboratory, No. 002726) or SOD1G37R (B6. Cg-Tg(SOD1*G37R) 1Dwc/J, JAX Laboratory, No. 016149). To acquire animals with and without SOD1G93A transgenic in the same litter as the optimal comparison, we crossed male SOD1G93A mice to non-transgenic genetic background strain C57BL/6J (JAX Laboratory, No. 000664). Pups were genotyped on the transgene SOD1G93A using primers specified by JAX Laboratory (oIMR0113, oIMR0114, oIMR7338 and oIMR7339) before weaning on day 21 postnatal. After genotyping, mice carrying SOD1G93A and their non-transgenic littermates were fed with special care approved for ALS disease study. Only males (unless specified) were selected and raised for the SRS imaging experiments unless particularly mentioned. Mice with SOD1G93A transgene were weighed every 3 days after 8 weeks postnatal to monitor the onset of muscle atrophy. Normally, SOD1G93A mice start to lose weight varying from 12 to 14 weeks. ALS behavioural staging monitor was performed starting from 12 weeks postnatal, with regular diet and hydrogel provided in the cages. SOD1G93A mice under ALS level 2 were monitored every 3 days, while mice over ALS level 3 were monitored daily until sacrifice for imaging or histology. Unless mentioned specifically, male SOD1G93A animals were subjected to *ex vivo* and *in vivo* SRS imaging for this study to reduce the variability resulted from a population of mixed genders. Copy number quantitative PCR tests were performed based on JAX Laboratory protocols. The FUSP525L and AAV-C9ORF72 mouse model of ALS has been described previously[5,41]. All experimental protocols and procedures were approved by the Institutional Animal Care and Use Committee of Harvard University. These procedures were consulted and supervised by the veterinarian of the Office of Animal Resources.

***Ex vivo* SRS imaging.** Mice carrying SOD1G93A and age-matched non-transgenic WT littermates were imaged at the age of 4, 8, 12 and 16 weeks postnatal. Mice were perfused with 4% paraformaldehyde, and their sciatic nerves were dissected from both legs. The sciatic nerves were rinsed in PBS and mounted on glass slides covered by a coverslip for SRS imaging. As described in the main text, we mainly imaged the midpoint of the dissected sciatic nerve (from the ventral root and ending at the branching point of tibial nerve).

The imaging was performed on an Olympus BX61WI upright microscope with FV300 scanning unit. The laser source employed was pico EMERALD from A.P.E., which provided a combined beam of two 7-ps lasers, whose wavelengths were 1,063 nm for Stokes beam and a tunable output for pump beam respectively. In this study, we focus on the CH high wavenumber region ($2,830–3,020\,cm^{-1}$) for imaging and hyperspectral imaging. For morphology imaging, the energy difference of the photons from these two wavelengths was around $2,850\,cm^{-1}$ which corresponded to the $CH_2$ symmetric stretching vibration in lipids (Supplementary Fig. 4b). The 1,063 nm laser was modulated by an electro-optical modulator at 10 MHz. The lasers were transmitted to scanning microscope and scanned by x–y galvo scanners and focused on the sample by a high NA objective. The objective we applied in the experiment was the IR version Olympus UPLANAPO 60X water immersion objective (NA 1.2). The transmitted light through the sample was collected by a Nikon oil immersion condenser (NA 1.4) and filtered by a band pass filter (Chroma, coherent anti-Stokes Raman scattering 890/220) to remove the 1,064 nm laser component. The pump laser was subsequently illuminated on a large area silicon photodiode detector (OSI Optoelectronics, S-100CL). The detected photocurrent was amplified by a filtered trans-impedance amplifier (Supplementary Fig. 20) and sent to the lock-in amplifier for demodulation. The demodulated signal was then sent to the data acquisition unit of the microscope and was eventually sent to a personal computer to show the images of the sample tissue. The optical power used on the sample is about 90 mW of pump and 80 mW of IR. The signal we get from myelin is about 110 μV with photodiode connected to a 50-ohm resistor. Our preamplifier will amplify this signal by about 23 times. However, the signal level depends on the thickness of sample, so it is varies depending on the samples preparation process. The imaging speed was 1 s/frame or 1.5 s/frame in a few occasions, and the image size was 512 by 512 pixels. The 3D reconstruction was generated by imageJ plug-in called volume viewer with all the three dimension scales corresponds to real life sizes.

**Whole-mount staining.** The whole mount staining was performed based on a previously published procedure[32]. After being fixed in 4% paraformaldehyde at 4 °C overnight, sciatic nerves were washed in 1% TritonX-100/PBS for 15 min. The nerve samples were then incubated in 10% donkey serum/1% TritonX-100/PBS at

4 °C overnight. On the next day, the nerves were incubated in anti-MBP (catalog number: MAB386MI, dilution factor: 1–500)/anti-Neurofilament (catalog number: SMI-32P, dilution factor: 1–500)/10% donkey serum/1% TritonX-100/PBS for 72 h at 4 °C. After incubation, the nerves were washed with 1% TritonX-100/PBS for 6 times (1 h each) at room temperature. The samples were then transferred into fluorescence-labelled secondary antibodies in 10% donkey serum/1% TritonX-100/PBS. After 48 h at 4 °C, the nerves were washed with 1% TritonX-100/PBS for 6 times (1 h each) at room temperature. For immunostaining of CD45, nerves were incubated in the fluorescence-labelled anti-CD45 (catalog number: BioLegend #103127 for 647 nm excitation and catalog number: BioLegend #103121 for 488 nm excitation, dilution factor: 1–200) in 10% donkey serum/1% TritonX-100/PBS for another 48 h and washed with 1% TritonX-100/PBS for 6 times (1 h each) at room temperature. High resolution images were then acquired based on LSM 880 confocal system at Harvard Center for Biological Imaging and our multiple channel SRS imaging separately. The SRS lipid imaging and two-photon excited epi-fluorescence imaging of MBP and neurofilament staining are performed at the same location by blocking different lasers. The MBP, which was excited at 543 nm, was imaged by two-photon excitation of 1,063 nm laser only under SRS microscope. The neurofilament, which was tagged at 488 nm, was imaged by two-photon excitation of pump laser at around 816 nm only under SRS microscope. SRS lipid imaging was obtained when both light beams were on the sample. The optical power used on the sample was about 90 mW of pump and 80 mW of IR. The imaging speed was 1 s/frame, and the image size was 512 by 512 pixels.

**Chemical composition analysis of lipid ovoids.** The nerve sample was imaged at 35 wavenumbers from 2,830 to 3,020 cm$^{-1}$. The first and last spectral point were not used for later spectral analysis. Seven chemicals were used for spectral retrieval. They are cholesterol(C), BSA, water and galactocerebroside, sphingomyelin, phosphatidylcholine, phosphatidylethanolamine. The least square method was used to retrieve the component of each chemical. We added the concentration of phosphatidylethanolamine and phosphatidylcholine because they have very similar spectrum. The lipid ovoids were manually selected and myelin sheath was selected from histogram from peak to 50% in the histogram. After point of interest was identified, the mean was then calculated for each chemical component. We imaged five locations on SOD1G93A mice and WT mice. The error bar was the standard deviation of mean in processed chemical content of five imaging locations. We used the one-tail unpaired student t-test to calculate the P value. The imaging parameter were the same as those used for ex vivo imaging experiment. The optical power used on the sample was about 90 mW of pump in the middle of scanning range, and the power was lower at both ends of spectral scanning range. The IR power was around 80 mW. The imaging speed was 1 s per frame, which means the image for each colour takes 1 s to take, and the image size was 512 by 512 pixels.

**Quantification of SRS images.** The commercialized image processing program ImageJ was used as a computational tool for SRS images quantification. The plug-in 'cvMatch_Template' was employed to calculate the correlation between SRS image and templates with the method called 'correlation coefficient'. The correlation pictures were then transformed to a threshold picture with an empirically chosen threshold value for each template. The areas with correlation coefficients higher than the threshold value would be designated as ROIs, which were emphasized by the red colour. Then the function of 'Analyze Particles' was used to select the ROIs satisfying defined roundness and size requirements. The circularity was assigned to be 0.1 smaller than the circularity obtained by template on the image from where the template had been selected. The size of the ROI was empirically selected, which was normally around 20% smaller than the size of original templates. The results of the 'Analyze Particles' function for each template were saved to a spreadsheet, and the total number was quantified. If the positions of the red areas counted by 'Analyze Particles' function from two templates overlapped, it would be counted for only once. 20 slices of z-stack images were analysed with this method, and the total number of lipid ovoids was calculated as the number of lipid ovoids of this z-stack image set. All sciatic nerve SRS images of ALS mouse models were quantified based on the same algorithm and template. Different lipid ovoids templates are used in human patients and mouse model quantifications.

Quantification of intact and degenerated axon numbers were performed from the 3D reconstruction of the z-stack images of sciatic nerves. The principle is the same as the one used for lipid ovoid counting. The difference was that we had 14 templates for intact myelinated nerve fibers and 4 templates for blocked nerve fibers. The ratio of blocked nerve fibers versus total myelinated nerve fibers (intact + blocked) was used to evaluate the progression of the disease.

We quantified the thickness of myelin sheath with longitudinal SRS image of the myelin sheath. We normally chose the image which sectioned the center of majority of myelin sheath. We utilized the plug-in 'tubeness' in FIJI to highlight the boundary of myelin sheath and calibrated the inside of myelin sheath and outside as zero intensity. Then we used the line profile of the image in the direction perpendicular the myelin sheath to calculate the thickness of each wall of myelin sheath. The thickness was defined by the width at the bottom of each myelin wall in the line profile. The position of line was scanned through whole image and the

thickness information of all myelin sheath at all locations in the image was output to a csv file for statistical analysis.

The quantification of human ALS versus control samples was principally the same as described by the ex vivo and in vivo SRS imaging of ALS mouse models based on ImageJ. But there are several differences. First of all, the templates were different from template used in mouse model. The templates for quantifying SRS images of human sample were selected from SRS images of nerves from ALS patients. They were tested to have good distinguishing ability between tissues from ALS patients and control. Second, there was very obvious loss of nerve fibers in the ventral nerves of ALS patients. Therefore, we also calculated the average signal intensity of each image as a quantification of averaged lipid enrichment. The quantification results from lipid ovoids counting were normalized by this averaged lipid signal intensity. So the final quantification had the unit of number of lipid ovoids of defined average lipid enrichment.

**Entire sciatic nerve scan.** The same optical pathway as the ex vivo SRS imaging was employed for entire sciatic nerve scan. Homemade software was used to automatically move the stage and take SRS images. The stage was moved by a step of 180 μm to image the whole sciatic nerve. This corresponded to 20% overlap between neighbor images, which was necessary for good stitching quality. The data processing initiated with compensating the light fall-off at image corners. The images were then stitched together with the FIJI plug-in 'Grid/collection stitching'[64]. The optical power used on the sample was about 90 mW of pump and 80 mW of IR. The imaging speed was 1 s per frame, and the image size was 512 by 512 pixels.

**In vivo end-point SRS imaging.** Mice carrying SOD1G93A and age-matched non-transgenic littermates were imaged at the age of 6, 7, 8, 12 and 16 weeks postnatal. The imaged mice were anaesthetized by 4% isoflurane (VetOne, catalog number: 502017) under an air flowing apparatus and kept alive during the entire imaging process. A small incision was made on the right leg of the imaged mouse to expose the sciatic nerve without removing any muscle. The mouse was mounted on a specially manufactured mouse rack (Fig. 3b), and the sciatic nerve was gently covered and protected by a coverslip on a metal holder (Fig. 3c). The in vivo imaging setup was similar to the ex vivo SRS imaging setup described above except that the back scattered light was collected. To effectively collect the reflected light, we installed a quarter wave plate and polarized beam splitter right before the microscope objective, so that the reflected lasers after passing the quarter wave plate a second time were switched to a polarization orthogonal to the input light and reflected by the polarized beam splitter. The light collection in the epi-mode was less efficient compared with transmission mode. Therefore, a higher laser power was employed for in vivo experiment. To be specific, the optical power used on the sample was about 120 mW of pump and 80 mW of IR. Mice in this section would be euthanized immediately after imaging, thus termed as in vivo end-point SRS imaging. Imaging speed was 1 s per frame, and the image size was 512 by 512 pixels.

**Long-term serial in vivo SRS imaging.** Mice carrying SOD1G93A and age-matched non-transgenic littermates were imaged using the in vivo end-point SRS imaging system described previously. However, all the imaged mice would be wound clipped on their incision immediately after imaging. On Day 7 after any surgery, the wound clips would be removed carefully during mouse anaesthetization. Long-term serial in vivo SRS imaging was initiated at the age of 5 weeks postnatal, with subsequent in vivo SRS imaging on the same mouse performed every 3 weeks until 11 weeks postnatal. Further behavioural monitoring and staging as described previously were performed to evaluate the diagnostic capability of SRS imaging. The optical power used on the sample was about 120 mW of pump and 80 mW of IR. The imaging speed was 1 s per frame, and the image size was 512 by 512 pixels.

**EMG.** To minimize electrical artifacts by the wound clips applied to the mice during SRS imaging, we conducted EMG first and subjected the mice for a recovery of 12 h before in vivo SRS imaging. Genotyped 5-week old mice were anaesthetized by 1.25% (v/v) avertin (Sigma T48402). Animals were placed immediately on a heating pad to maintain their core temperature at 37 °C. Measurements were performed using a Cardinal Synergy EMG machine. A ground self-adhesive gelled surface electrode was placed over the tail. Potentials were recorded from several sites of the hind-limb muscles with a concentric needle electrode (30 G) using a gain of 50 μV per division and a band pass filter with low and high cut-off frequency settings of 20 and 10,000 Hz, respectively. The entire recording process took 15–20 min per mouse[46].

**Minocycline treatment and serial in vivo SRS imaging of SOD1G93A mouse model of ALS.** To minimize the variation of SOD1G93A transgene copy number, we purchased 30 SOD1G93A mice at the age of 4 weeks from JAX Laboratory (No. 002726), where both low copy number and high copy number animals have been removed. These animals were supposed to carry 21–30 copies SOD1G93A transgene. An equal number of mice (N = 15) were randomly subjected to

minocycline (Sigma Aldrich, M9511) or vehicle (saline) treatment starting from 5 weeks of age, which is the first age point of *in vivo* SRS imaging. Unfortunately, two vehicle treated mice died accidentally due to the technical problem of isoflurane anaesthesia system during the first imaging. Therefore, 15 minocycline treated animals and 13 vehicle treated animals were then subjected to the course of the drug treatment and SRS imaging study. The minocycline treated animals were injected intraperitoneally daily at a concentration of 10 mg/kg animal body weight[39], while the vehicle group were treated with the same volume (100 μl) of saline. The three age points of *in vivo* imaging were 5, 8 and 11 weeks, which was the same as the long-term serial *in vivo* SRS imaging for untreated *SOD1*G93A animals versus controls. Each mouse was immediately injected with Meloxicam (1mg per 1kg mouse body weight) post-operation to reduce the inflammatory effects and pain. All mice after surgery would be wound-clipped on their incision for 7-day recovery and monitored closely based on the procedure as the described long-term serial *in vivo* SRS imaging. The imaging parameters were the same as those used in long term *in vivo* imaging. The optical power used on the sample was about 120 mW of pump and 80 mW of IR. The imaging speed was 1 s/frame, and the image size was 512 by 512 pixels.

**Sciatic nerve crush experiment.** *SOD1*G93A mice were raised as previously described. At the age of 5 weeks postnatal, *SOD1*G93A mice and their WT littermates were subjected to sciatic nerve crush. The animals were anaesthetized first by 1.25% (v/v) avertin (Sigma T48402), and a small incision on the right hind-limb was made to expose the sciatic nerve. Before the crush, a pair of fine forceps was clipped with black carbon powder (Strem Chemicals Inc., CAS No. 1333-86-4) to label the site of crush. The black carbon labelled forceps were then used to tightly crush the middle point of the exposed sciatic nerve for 3 times. Each time of crush last for 10 and 5 s interval was between each crush. After the sciatic nerve crush, the incision of the animals was sutured by wound clips. They were returned to cage and raised until the time of SRS imaging. If they were not sacrificed on Day 7 after sciatic nerve crush, the wound clips would be removed carefully during mouse anaesthetization.

**EAE model of MS.** Female SJL/J mice (JAX Laboratory, No. 000686) at the age of 8 to 9 weeks were injected subcutaneously in each axillary areas with PLP 139–151 peptide (100 μg/mouse), together with 400 μg Mycobacterium tuberculosis H37 RA (DIFCO Laboratories) in an emulsion containing equal parts of PBS and IFA and PBS (Sigma-Aldrich). Pertussis toxin, 500 ng (List Biological Laboratories) was injected i.p. on the day of the immunization and on day 2 post-immunization. Mice were scored daily for clinical signs of EAE on a scale from 1 to 5 according to the severity of the disease. Scoring of the clinical symptoms as follows: 0 = no clinical signs; 0.5 = tail limpness; 1 = complete tail limpness, loss of bladder control and ruffling of fur; 2 = tail limpness and moderate hind limb weakness or unsteady gait; 3 = complete hind limb paralysis; 4 = hind limb paralysis and fore limb paralysis combined with cachexia (less than 25% of body weight); 5 = moribund.

**SRS imaging of human samples.** Formalin fixed post-mortem human ventral or dorsal root samples (ALS patients versus age-matched controls without a history of neurodegenerative disease) were obtained at Massachusetts General Hospital using a Partners IRB protocol approved by Partners Human Research Committee. The setup used to visualize human ventral or dorsal root samples was the same as mouse *ex vivo* SRS imaging. Fixed human ventral or dorsal root samples were first washed with PBS and then mounted to glass slide with a coverslip embedded by PBS. Imaging speed was 1.5 s/frame, and the image size was 512 by 512 pixels.

**Data availability.** The authors declare that the majority of data supporting the findings of this study are available within the article and the Supplementary Materials. Additional data that support the findings of our study are available from the corresponding authors on request.

**Statistical analysis.** Statistical analysis was performed with data acquired for at least four independent repeats. Student's (unpaired) *t*-test was used as a criterion to evaluate the differences between *SOD1*G93A and WT non-transgenic mice. $P < 0.05$ was considered as the significance cut-off.

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

## Acknowledgements

We thank P. Arlotta, A. Cohen, F. Engert and J. Lichtman for informative suggestions on this work. We also thank A. Huang for the discussion on spectral reconstruction and S. Oh for the help on SRS signal measurement. This work was supported by the Howard Hughes Medical Institute, Target ALS, NINDS grant RO1NS01NS089742 to KE and National Institute of Health/National Institute of Biomedical Imaging and Bioengineering (Award #: 5R01EB010244). D.A.M. was supported by the Massachusetts Alzheimer's Disease Research Center (NIA P50 AG005134) and NCI 5T32CA009216-34. N.S. was a 2011 Lilly Scientific Fellow and received grant from The Mochida Memorial Foundation for Medical and Pharmaceutical Research. N.A.S and A.S. were supported by the National Institute of Neurological Disorders and Stroke (NINDS) (Award#: R01NS07377). This work was supported by the National Institutes of Health/National Institute on Aging (P50AG016574 (L.P.)), National Institutes of Health/National Institute of Neurological Disorders and Stroke (R21NS084528 (L.P.), R01NS088689 (L.P.), R01NS063964 (L.P.); R01NS077402 (L.P.); P01NS084974 (L.P.)), National Institute of Environmental Health Services (R01ES20395 (L.P.)), Mayo Clinic Foundation (L.P.), Mayo Graduate School (J.C.), ALS Association (L.P.) and Robert Packard Center for ALS Research at Johns Hopkins (L.P.), Target ALS (L.P.).

## Author contributions

K.E., X.S.X., F.T. and W.Y. developed the hypothesis and designed the study. W.Y. designed the filtered trans-impedance amplifier circuit for photodiode and performed image processing, imaging tiling and 3-D imaging reconstruction. W.Y. and F.T. performed the *ex vivo* and *in vivo* SRS imaging in *SOD1*G93A mouse model of ALS with the contribution from S.S.H. (comparative histology), F.L., M.J. and W.Y. designed and made the animal rack and optical setup for *in vivo* imaging. M.J. developed the SRS tiling software. W.Y. designed software algorithms to performed the hyperspectral SRS imaging and chemical decomposition. S.S.-U. and N.S. (long-term serial SRS imaging) and Y.L. (entire sciatic nerve scan). J.W. and F.T. designed and conducted the characterization of lipid ovoid by whole mount staining and immunohistochemistry, and W.Y. performed SRS and two-photon excited fluorescence imaging on those samples. W.Y., F.T. and J.M. conducted the serial survival surgery and *in vivo* SRS imaging of *SOD1*G93A mice under minocycline treatment. W.Y., F.T. and D.A.M. performed the SRS imaging of nerve sample from postmortem ALS patients or controls. J.S.S. and F.T. performed the comparative study between SRS imaging and EMG with contribution from W.Y. to taking SRS images and K.E. to the experimental design. F.T. and R.Z. prepared the samples in the sciatic nerve crush experiment and W.Y. performed SRS imaging on the samples with contribution from K.E. to the experimental design. J.C. and L.P. prepared and the *ex vivo* sciatic nerve samples for AAV-*C9ORF72* mouse ALS model. A.S. and N.A.S. prepared and the *ex vivo* sciatic nerve samples for *FUS*P525L mouse ALS model. E.L.-D. and J.S. prepared and offered the EAE model mice. F.T. prepared and offered the *SOD1*G37R mouse ALS model. W.Y. developed the quantification algorithms of axon cross-section counting, *ex vivo* and *in vivo* lipid ovoid counting, myelin thickness, axon size counting and lipid ovoid chemical composition analysis with input and suggestions from K.E., F.T., R.Z. and X.S.X. for design and optimization. F. T.,W.Y., D.A.M., R.Z., X.S.X. and K.E. wrote the manuscript with revision and input from all the authors.
