## [Peer Review File · Nature Communications]

Reviewers' comments:

Reviewer #1 (Remarks to the Author):

This manuscript presents the application of stimulated Raman scattering (SRS) imaging tools to visualize peripheral degeneration in several mouse models of ALS and human postmortem tissue. Based on the quantitative morphological analysis of predominantly oval lipid-enriched structures in a series of SRS images, the authors demonstrate that disease progression could be followed successfully over time and that the sensitivity of first detectable signs of peripheral nerve degeneration by SRS coincides with those of muscle denervation by electromyography (EMG). While I do not consider the presented work as a contribution to the further technological development of SRS imaging, i.e. a well-established and previously well-reported setup for SRS imaging was used, the originality of this work rather counts on the application of SRS imaging for the systematic studies of peripheral nerve degeneration and its cross-evaluation with conventional diagnostic methods. To the best of my knowledge, the latter has not been reported yet. Therefore, this work may be greatly appreciated by the community working in neurodegenerative disease research.

The experimental work, for the most part, was thoroughly performed. The manuscript was clearly written and is well organized. I recommend publication in Nature Communications after the authors will have addressed the following minor issues in a revised manuscript:

- 1) Introduction, p.3, and Discussion, p.22: In referring to the original literature that first reported SRS imaging and demonstrated the linear dependence of SRS image contrast on the concentration of vibrational modes in the sample of interest, the authors may reconsider citing all pioneering work on SRS imaging, i.e. also giving appropriate credit to the work reported by other research teams.
- 2) Results, p.5+9, and Supp. Fig. 4b: Since the main results of this work are predominantly based on the morphological analysis of the oval lipid-enriched structures, it would be highly desirable if the authors could provide a more detailed characterization of these structures! By exploiting both the 3D imaging and the multi-spectral imaging capabilities of their SRS microscope setup, the authors can provide information about the volume and chemical composition of a typical "ovoid", respectively. For example, on p.9, the authors refer to "... counting planar lipid ovoids ...". Do the observed ovoids really resemble planar structures? Regarding the relative chemical composition of proteins and lipids of a typical ovoid, the authors use the term "lipid-enriched" without providing spectral evidence. In fact, the only SRS spectra shown in Supplementary Fig. 4b are those for the lipid-rich myelin fiber and for the protein-rich axon inside. Please amend by showing at least an SRS spectrum (or alternatively a spontaneous Raman spectrum) recorded inside an ovoid: What are the spectral differences between ovoid, myelin sheet, and its inside within the CH stretching region exploited in this work?
- 3) Methods, p.24-27, and all Figs. showing SRS images and spectra: Unfortunately, experimental details regarding the actual pump and Stokes beam powers at the sample used for the different experiments are missing! Furthermore, experimental details regarding the absolute amplitudes and units of the processed electronic SRS signals are missing! Please amend power values, calibrated look-up tables (LUT's) for SRS images and signal levels for SRS spectra. In addition, please provide details regarding the multi-spectral SRS imaging shown in Supp. Fig. 4b, such for example the acquisition time per spectral scan step?

Reviewer #2 (Remarks to the Author):

In this manuscript, Tian and colleagues demonstrate how SRS microscopy can be used to image myelin and demyelination in peripheral nerves both from animal models and human specimens. Their

scientific methods are sound and the breadth of experiments they have executed is impressive. They claim that SRS microscopy can be used to detect early peripheral nerve degeneration in ALS and, consequently, SRS can be used to study nerve degeneration in ALS models and to both diagnose and monitor disease progression in ALS patients.

I agree with the authors' claim that a highly sensitive method for detecting early degenerative changes in ALS models could be useful in "detecting the effects of experimental interventions." But I also wonder why they have not shown how SRS could measure the effects of such interventions (e.g. forced expression of ATF3 in SOD1G93A mice).

I do not agree with the authors' claim that SRS microscopy could ultimately serve as a means for clinical diagnosis and monitoring in ALS patients. While there have been important technologic advances in the translation of SRS microscopy out of the Xie group, it is likely that in vivo epi or reflectance SRS imaging hardware would be too large to deliver through a needle. The use of a handheld reflectance mode SRS probe would require surgical exposure- and an incision or several centimeters to image involved motor nerves. Moreover, it is likely that there would be a compromise in image quality in a handheld SRS system in comparison to the ideal imaging conditions that can be accomplished in the laboratory setting.

Furthermore, the authors suggest that the added benefit of SRS microscopy over EMG would be the added structural information that SRS can deliver. However, it is not clear whether SRS-detectable changes in affected nerves could be used to differentiate between types of axonopathies, therefore limiting the capacity of SRS as a diagnostic modality. For example, the histoarchitectural changes in ALS models and the models of crush injuries shown here seem similar.

In a revision, I would recommend that the authors focus on SRS microscopy as a way of studying demyelination as a research method rather than a possible clinical imaging modality.

Minor comments:

Was there any correlation between degree of lipid ovoid formation with degree of motor neuron loss in the spinal cord of ALS animals? Showing that the proportion of degenerated sciatic nerve correlates with the proportion of motor neuron loss in lumbar spine at the same time points would strengthen the claim that this was due to the disease process affecting motor neurons. Alternatively, according to the timeline in figure 6e, central motor neuron changes occur later on (~10 weeks), so do they suggest that degenerative changes are detectable in the axon before any changes in the motor neuron cell body?

As the authors mention, the clinical manifestation of ALS can be quite varied. Therefore, some clinical background on the ALS human patients in which specimens in this paper were derived would be good to include. Were these ALS patients limb-onset ALS (as opposed to bulbar-onset)? And did nerve roots imaged correspond to symptomatic limbs. Did any patients have familial forms of ALS/FTLD or were they all sporadic?

Reviewer #3 (Remarks to the Author):

NCOMMS-16-01992

"Monitoring peripheral nerve degeneration in ALS cases and mouse models by label-free Stimulated Raman Scattering imaging", by Tian and colleagues

This manuscript by Tian and colleagues developed a label-free, highly sensitive imaging device based

on Stimulated Raman Scattering (SRS) microscopy to detect the degeneration of peripheral nerves in mouse models of ALS, peripheral nerve pathologies induced by mechanical injuries or demyelinating agents, and human ALS tissues. The authors present stunningly high quality images supporting the utility and sensitivity of SRS microscopy in the diagnosis and monitor of neurodegeneration in the peripheral nerves. Their results make a compelling case in favor of SRS microscopy as an important diagnostic and research tool for peripheral nerve degeneration in ALS and other injury paradigms. While this manuscript contains many novel findings, there are a few areas that require major attention in order to improve the clarity and presentation of the manuscript.

1. Although the images from SRS microscopy are of very high quality with clear delineation of the myelin sheath and the node of Ranvier in control peripheral nerves, the specificity of these images and their correlations with the underlying pathology in diseased tissues remain poorly characterized. For instance, the authors described abnormalities in the ex vivo sciatic nerves from SOD1G93A mice as "oval lipid-enriched structures" and called it as "lipid ovoids" (Figure 1d-g). But, what are the actual contents within these structures, and where is the evidence that these lesions are indeed enriched in lipids? This is particularly puzzling given that SOD1G93A mutation has been shown to cause axonal transport, rather than demyelinating, defects. There are at least two possibilities to account for the "lipid ovoids". First, they reflect the presence of disorganized axonal proteins, such as neurofilaments. Second, they may represent disorganized myelin sheaths that cover areas of axonal swellings. It is important to present histopathology examinations, including special stains for axonal proteins and myelin sheaths, to make a strong correlation between SRS microscopic imaging and histopathology. Similar histopathology images should be provided for nerve crush and EAE models because the histopathology changes in these models are likely to be entirely different from SOD1G93A.

2. Another major issue is that the contents and the sequence of data presentation in this manuscript are quite disorganized. There are a few areas that can be improved to provide clarity and sound logics. First, the sequence of data presentation in Figure 1 should be re-organized. In its current format, the text describing in Figure 1 is out of order (pp. 5-6). For instance, Figure 1 d-g are cited before Figure 1a-c. Second, as the title indicates, the primary goal of this manuscript is to demonstrate the utility of SRS microscopy in the detection and monitoring of peripheral nerve degeneration in ALS models and ALS human tissues. As such, the data on nerve crush and EAE models should be relocated at the very end of this manuscript, and serve as extension of SRS microscopy to other models of peripheral neuropathy.

3. There are many redundant and unnecessary image panels in the formal figures and supplemental figures. For instance, there are too many "control" SRS microscopic images, e.g. 4, 8, 12 and 16 weeks in Figure 1, all look almost identical. Similar repetitive panels are also present in Figures 2 and 4. In addition, Supplementary Figure 3 is completely redundant and can be incorporated into Figure 1, Supplementary Figure 9 is redundant and should be removed, Supplementary Figure 11 contains information identical to Figure 1, etc, etc. Finally, I am not sure about the contribution from Supplementary Figure 17. For a biology-heavy manuscript, information in this figure is definitively redundant.

4. Supplementary Figure 4 show results measuring the lipid and protein density that correlates with SRS microscopic signal intensity. However, this figure is never cited in the main text, and the results are never explained at all. I believe the data in this figure are important and should be included in Figure 1 along with some histopathology images to establish the correlation between SRS images, biochemical quantification of proteins and lipids, and histopathology characteristics of these lesions in SOD1G93A, nerve crush and EAE model.

Point-by-point response to the referees' comments

Reviewer #1 (Remarks to the Author):

This manuscript presents the application of stimulated Raman scattering (SRS) imaging tools to visualize peripheral degeneration in several mouse models of ALS and human postmortem tissue. Based on the quantitative morphological analysis of predominantly oval lipid-enriched structures in a series of SRS images, the authors demonstrate that disease progression could be followed successfully over time and that the sensitivity of first detectable signs of peripheral nerve degeneration by SRS coincides with those of muscle denervation by electromyography (EMG). While I do not consider the presented work as a contribution to the further technological development of SRS imaging, i.e. a well-established and previously well-reported setup for SRS imaging was used, the originality of this work rather counts on the application of SRS imaging for the systematic studies of peripheral nerve degeneration and its cross-evaluation with conventional diagnostic methods. To the best of my knowledge, the latter has not been reported yet. Therefore, this work may be greatly appreciated by the community working in neurodegenerative disease research.

Our response:

Thank you for your positive comments concerning of our work. We agree with Reviewer #1 that our study is more about the application of SRS imaging in ALS models rather than the innovation of new imaging techniques. We apologize for this misconception created by our first draft of the manuscript. Like the referee, we hope that the publication of this manuscript concerning the use of SRS imaging can facilitate the measure of neurodegenerative processes, especially in mouse models and postmortem tissues.

Reviewer #1 (Remarks to the Author):

The experimental work, for the most part, was thoroughly performed. The manuscript was clearly written and is well organized. I recommend publication in Nature Communications after the authors will have addressed the following minor issues in a revised manuscript:

1) Introduction, p.3, and Discussion, p.22: In referring to the original literature that first reported SRS imaging and demonstrated the linear dependence of SRS image contrast on the concentration of vibrational modes in the sample of interest, the authors may reconsider citing all pioneering work on SRS imaging, i.e. also giving appropriate credit to the work reported by other research teams.

Our response:

We sincerely appreciate your positive comments on the writing and organization of our manuscript, and we regret not more-widely citing important and pioneering studies on

SRS. To correct this oversight, we have now added the following references to our manuscript at locations that we hope the reviewer will find appropriate:

Dake, F., Ozeki, Y. & Itoh, K. Principle confirmation of stimulated Raman scattering microscopy. Optics & Photonics Japan (OPJ2008), paper 5pC12, Tsukuba (2008).

Nandakumar, P., Kovalev, A. & Volkmer, A. Vibrational imaging based on stimulated Raman scattering microscopy. New Journal of Physics 11, 033026 (2009).

Ozeki, Y., Dake, F., Kajiyama, S.i., Fukui, K. & Itoh, K. Analysis and experimental assessment of the sensitivity of stimulated Raman scattering microscopy. Opt Express 17, 3651-3658 (2009).

Reviewer #1 (Remarks to the Author):

2) Results, p.5+9, and Supp. Fig. 4b: Since the main results of this work are predominantly based on the morphological analysis of the oval lipid-enriched structures, it would be highly desirable if the authors could provide a more detailed characterization of these structures! By exploiting both the 3D imaging and the multi-spectral imaging capabilities of their SRS microscope setup, the authors can provide information about the volume and chemical composition of a typical "ovoid", respectively. For example, on p.9, the authors refer to " ... counting planar lipid ovoids ...". Do the observed ovoids really resemble planar structures?

Our response:

In the updated version of the manuscript an expanded chemical composition analysis by performing a more detailed SRS spectrum scan reveals that the major composition of "lipid ovoids" are lipids, likely derived from myelinating cells (**Fig. 2b**). Our new immunostaining of the neuronal antigen (neurofilament or NF) and blood cell antigen (CD45) indicates that lipid ovoids were not infiltrating immune cells and were largely devoid of neuronal structural proteins (**Fig. 2a**). By immunostaining some, but not all lipid ovoids displayed co-localization with MBP (myelin basic protein) staining, supporting the notion that they have a myelinating cell origin. We note that these later experiments are difficult to carry out without artifact. The fixation techniques used for immunofluorescence include necessary treatments with detergents, which reduce the lipid signal in SRS imaging. Alternatively, imaging with SRS and then fixing and staining can lead to difficulties in precise co-localization. However, we hope that the reviewer feels that our findings, within these technical bounds, are sufficient to support our conclusion that these ovoid features we quantify are reasonably termed "lipid ovoids" and that they are likely derived from myelinating cells. (**Supplementary Fig. 6**). We additionally provide data that indicate that "lipid ovoids" are unlikely to be phagocytic cells engulfing lipids (**Fig. 2**).

Furthermore, at Reviewer #1's suggestion, we performed the three dimensional (3-D) reconstruction of typical SRS lipid imaging on sciatic nerve of wild-type (WT) and

SODIG93A mice. (**Fig. 1f**). The reconstructed 3-D images allow readers to have a direct impression of the size, shape and locations of these lipid ovoids. The width of these ovoids was normally less than 15 μm , which was slightly smaller than that of the nearby myelin sheathes. The length of lipid ovoids varied dramatically from 15 to 40 μm . Some lipid ovoids seemed to contain structures inside, while some were empty. We believe the term of lipid ovoids should be warranted based on our data overall

Reviewer #1 (Remarks to the Author):

Regarding the relative chemical composition of proteins and lipids of a typical ovoid, the authors use the term "lipid-enriched" without providing spectral evidence. In fact, the only SRS spectra shown in Supplementary Fig. 4b are those for the lipid-rich myelin fiber and for the protein-rich axon inside. Please amend by showing at least an SRS spectrum (or alternatively a spontaneous Raman spectrum) recorded inside an ovoid: What are the spectral differences between ovoid, myelin sheet, and its inside within the CH stretching region exploited in this work?

Our response:

We agree with Reviewer #1 that a more detailed spectrum analysis could add substantial value to our manuscript. Therefore, we performed the SRS spectrum scan and chemical composition study on 1) WT myelin, 2) *SODIG93A* myelin and 3) *SODIG93A* lipid ovoids. We reported the result of the chemical decomposition in **Fig. 2**. We have presented the spectra of chemicals in **Supplementary Fig. 4b**, in which we also presented the typical spectrum of myelin in wild type and *SODIG93A* mice as well as lipid ovoids in *SODIG93A* mice.

Reviewer #1 (Remarks to the Author):

3) Methods, p.24-27, and all Figs. showing SRS images and spectra: Unfortunately, experimental details regarding the actual pump and Stokes beam powers at the sample used for the different experiments are missing! Furthermore, experimental details regarding the absolute amplitudes and units of the processed electronic SRS signals are missing! Please amend power values, calibrated look-up tables (LUT's) for SRS images and signal levels for SRS spectra. In addition, please provide details regarding the multi-spectral SRS imaging shown in Supp. Fig. 4b, such for example the acquisition time per spectral scan step?

Our response:

Thank you for the comment. We regret that we did not include enough information concerning methods for the imaging. For *ex vivo* imaging, we used 90 mW of pump laser and 80 mW of Stokes laser on the sample. We used an amplified diode with amplification of 23 times. With this diode we obtained OA's signal at 13 mVs. So without amplification the signal would be 565 μVs on a 50 Ohm diode at 2850 cm^{-1} . The signal of myelin was at about 110 μVs . This is a typical value, since the signal level for *ex vivo*

imaging depends on the thickness of nerve. For in vivo imaging, the signal levels further depend on the sample surface conditions. We used imageJ's 'red hot' as LUT for the representative SRS images in the main figures. The acquisition time was 1 second per frame with resolution of 512 by 512 for in vivo, ex vivo imaging and spectral scanning. Standard chemicals for spectra decomposition: phosphatidylcholine (P3556-100MG), phosphatidylethanolamine (P7943-5MG), sphingomyelin (S0756-50MG) and galactocerebroside (C4905-25MG) were obtained from Sigma Aldrich. Bovine serum albumin (BSA) was dissolved in water with 10% of concentration by weight. We have updated the materials and methods with this information.

Reviewer #2 (Remarks to the Author):

In this manuscript, Tian and colleagues demonstrate how SRS microscopy can be used to image myelin and demyelination in peripheral nerves both from animal models and human specimens. Their scientific methods are sound and the breadth of experiments they have executed is impressive. They claim that SRS microscopy can be used to detect early peripheral nerve degeneration in ALS and, consequently, SRS can be used to study nerve degeneration in ALS models and to both diagnose and monitor disease progression in ALS patients.

I agree with the authors' claim that a highly sensitive method for detecting early degenerative changes in ALS models could be useful in "detecting the effects of experimental interventions." But I also wonder why they have not shown how SRS could measure the effects of such interventions (eg forced expression of ATF3 in *SOD1G93A* mice).

Our response:

We thank the referee for their overall positive impression of our manuscript. We also agree that the sensitivity and possibility of early detection are the strength of SRS imaging. Although it was a major undertaking, we decided to test our ability to put SRS imaging to work in evaluating interventions in ALS mouse models. Although ATF3 overexpression would have been an interesting intervention, these mice were not immediately available to us. Instead, we opted to investigate the experimental therapeutic minocycline, which is a drug previously reported to slow disease in the *SOD1G93A* and *SOD1G37R* mouse model of ALS, but which failed to meet therapeutic endpoints in clinical trial. We felt that SRS imaging might be a useful means to see if the positive effects of minocycline could be replicated in mouse models. This would provide greater confidence in minocycline's efficacy in mice, leading to a stronger conclusion that lack of this compounds effects in patients was either due to a difference in mouse and human physiology or an *SOD1* specific effect that could be subsequently investigated.

We therefore initiated drug and vehicle treatment in animals at 5 weeks of age, the earliest time point we could begin to detect degenerative effects of the *SOD1G93A* transgene. Our results indeed suggested that SRS imaging is able to detect the difference in lipid ovoid deposition and peripheral nerve degeneration between minocycline treated

and vehicle treated animals by 8 weeks of age (the third week of minocycline treatment), when both groups were otherwise pre-symptomatic (**Fig. 5**). This effect was sustained at 11 weeks as well. We also found that the rich quantitative and single animal nature of SRS imaging data also allowed us to correlate degeneration with both drug effect and SOD1 copy number, providing further confidence in our finding. We believe and hope the reviewer agrees that our minocycline treatment experiment has convincingly demonstrated that SRS imaging can indeed be used for the evaluation of future ALS interventions.

Reviewer #2 (Remarks to the Author):

I do not agree with the authors' claim that SRS microscopy could ultimately serve as a means for clinical diagnosis and monitoring in ALS patients. While there have been important technologic advances in the translation of SRS microscopy out of the Xie group, it is likely that in vivo epi or reflectance SRS imaging hardware would be too large to deliver through a needle. The use of a handheld reflectance mode SRS probe would require surgical exposure- and an incision or several centimeters to image involved motor nerves. Moreover, it is likely that there would be a compromise in image quality in a handheld SRS system in comparison to the ideal imaging conditions that can be accomplished in the laboratory setting.

Our response:

While we are hopeful that advances in technology might one day bring SRS imaging to the clinic, we accept the reviewer's criticism that this day may not be immediately at hand. Therefore, we have made corresponding revisions in the discussion of the application of SRS imaging. Nevertheless, we still believe that SRS imaging might have potential usage in certain situations such as the neural stem cell transplantation therapy of ALS, where spinal cord exposure is mandatory procedure. Additionally, SRS imaging of pathological tissue might in the future allow for rapid analysis of postmortem nerve tissue. Nevertheless, to reflect the reviewers concerns and as it is not the central point of our manuscript, we have greatly revised the discussion to acknowledge these difficulties.

Reviewer #2 (Remarks to the Author):

Furthermore, the authors suggest that the added benefit of SRS microscopy over EMG would be the added structural information that SRS can deliver. However, it is not clear whether SRS-detectable changes in affected nerves could be used to differentiate between types of axonopathies, therefore limiting the capacity of SRS as a diagnostic modality. For example, the histoarchitectural changes in ALS models and the models of crush injuries shown here seem similar.

Our response:

We accept the referees' critiques here as well and have substantially revised our discussion on diagnostic use to reflect these reasonable points.

Reviewer #2 (Remarks to the Author):

In a revision, I would recommend that the authors focus on SRS microscopy as a way of studying demyelination as a research method rather than a possible clinical imaging modality.

Our response:

We have modified the related paragraphs in the discussion section accordingly.

Reviewer #2 (Remarks to the Author):

Minor comments:

Was there any correlation between degree of lipid ovoid formation with degree of motor neuron loss in the spinal cord of ALS animals? Showing that the proportion of degenerated sciatic nerve correlates with the proportion of motor neuron loss in lumbar spine at the same time points would strengthen the claim that this was due to the disease process affecting motor neurons. Alternatively, according to the timeline in figure 6e, central motor neuron changes occur later on (~10 weeks), so do they suggest that degenerative changes are detectable in the axon before any changes in the motor neuron cell body?

Our response:

We do indeed believe that we are detecting very early peripheral changes in the nerve using SRS that correlate strongly with other signs of neural degeneration. The changes in EMG that are occurring within a similar time scale support this view. Certainly the early changes we report here (5-8 weeks) are likely substantially before the loss of motor neurons, or certainly before one could easily detect a change in their number by standard motor neuron counting methods. In previous **Fig. 6e**, which is now **Fig. 8e**, we summarize how the occurrence of lipid ovoids fits in temporally with events in the model. The sensitivity of SRS imaging likely in large part derives from the very large area of peripheral nerve, that when initially fragmenting can give rise to many lipid ovoids. This provides a large signal that can be detected early. Based on historic counts in our lab and as well as others, we believe it is not readily possible to detect a loss in motor neuron cell bodies at this 5-8 week window. Although we did not correlate the number of lipid ovoids with the number of motor neurons per se, we have correlated the number of lipid ovoids with the copy number of mutant *SOD1*. There was indeed a strong correlation, further supporting that the lipid ovoids are a measure of the action of the mutant transgene.

Reviewer #2 (Remarks to the Author):

As the authors mention, the clinical manifestation of ALS can be quite varied. Therefore, some clinical background on the ALS human patients in which specimens in this paper

were derived would be good to include. Were these ALS patients limb-onset ALS (as opposed to bulbar-onset)? And did nerve roots imaged correspond to symptomatic limbs. Did any patients have familial forms of ALS/FTLD or were they all sporadic?

Our response:

These ALS patients all suffered from limb-onset ALS, and they were all sporadic cases. This information has been added to the revised manuscript.

Reviewer #3 (Remarks to the Author):

NCOMMS-16-01992

"Monitoring peripheral nerve degeneration in ALS cases and mouse models by label-free Stimulated Raman Scattering imaging", by Tian and colleagues

This manuscript by Tian and colleagues developed a label-free, highly sensitive imaging device based on Stimulated Raman Scattering (SRS) microscopy to detect the degeneration of peripheral nerves in mouse models of ALS, peripheral nerve pathologies induced by mechanical injuries or demyelinating agents, and human ALS tissues. The authors present stunningly high quality images supporting the utility and sensitivity of SRS microscopy in the diagnosis and monitor of neurodegeneration in the peripheral nerves. Their results make a compelling case in favor of SRS microscopy as an important diagnostic and research tool for peripheral nerve degeneration in ALS and other injury paradigms. While this manuscript contains many novel findings, there are a few areas that require major attention in order to improve the clarity and presentation of the manuscript.

Our response:

We thank the reviewer for their positive comments.

Reviewer #3 (Remarks to the Author):

1. Although the images from SRS microscopy are of very high quality with clear delineation of the myelin sheath and the node of Ranvier in control peripheral nerves, the specificity of these images and their correlations with the underlying pathology in diseased tissues remain poorly characterized. For instance, the authors described abnormalities in the ex vivo sciatic nerves from SOD1G93A mice as "oval lipid-enriched structures" and called it as "lipid ovoids" (Figure 1d-g). But, what are the actual contents within these structures, and where is the evidence that these lesions are indeed enriched in lipids? This is particularly puzzling given that SOD1G93A mutation has been shown to cause axonal transport, rather than demyelinating, defects. There are at least two possibilities to account for the "lipid ovoids". First, they reflect the presence of disorganized axonal proteins, such as neurofilaments. Second, they may represent disorganized myelin sheaths that cover areas of axonal swellings. It is important to present histopathology examinations, including special stains for axonal proteins and

myelin sheaths, to make a strong correlation between SRS microscopic imaging and histopathology. Similar histopathology images should be provided for nerve crush and EAE models because the histopathology changes in these models are likely to be entirely different from SOD1G93A.

Our response:

Thank you for your comments. We addressed this question partially in the response to Point (2) from Reviewer #1. We have performed detailed analysis of chemical composition and immunohistochemistry as described above. The new chemical composition study by a more detailed SRS spectrum scan suggested that lipid ovoids resemble the spectrum of chemical components derived from myelin and were rich in lipids (**Fig. 2b**). Further immunofluorescence studies based on whole mount stain of sciatic nerves demonstrate that they were not likely to be immune cells or primarily from an axonal origin (**Fig. 2b** and **Supplementary Fig. 17**). Instead, lipid ovoids were most likely to be composed of myelin lipids and in some cases could be clearly stained with anti-MBP antibodies. We also now show a 3-D reconstruction of sciatic nerves from *SOD1G93A* and WT mice based on SRS imaging. These 3-D images clearly show the typical size, shape and location of lipid ovoids (**Fig. 1f**).

Reviewer #3 (Remarks to the Author):

2. Another major issue is that the contents and the sequence of data presentation in this manuscript are quite disorganized. There are a few areas that can be improved to provide clarity and sound logics. First, the sequence of data presentation in Figure 1 should be re-organized. In its current format, the text describing in Figure 1 is out of order (pp. 5-6). For instance, Figure 1 d-g are cited before Figure 1a-c. Second, as the title indicates, the primary goal of this manuscript is to demonstrate the utility of SRS microscopy in the detection and monitoring of peripheral nerve degeneration in ALS models and ALS human tissues. As such, the data on nerve crush and EAE models should be relocated at the very end of this manuscript, and serve as extension of SRS microscopy to other models of peripheral neuropathy.

Our response:

We appreciate this clear feedback. In response, we have reorganized the previous **Fig. 1** and re-ordered display items so that the logic of how lipid ovoids in *ex vivo* sciatic nerve SRS images have been better illustrated (**Fig. 1**). Second, we have relocated the EAE, crush model and human experiments as suggested. (**Fig. 7**)(**Fig. 8e**).

Reviewer #3 (Remarks to the Author):

3. There are many redundant and unnecessary image panels in the formal figures and supplemental figures. For instance, there are too many "control" SRS microscopic images, e.g. 4, 8, 12 and 16 weeks in Figure 1, all look almost identical. Similar repetitive panels are also present in Figures 2 and 4. In addition, Supplementary Figure 3 is completely

redundant and can be incorporated into Figure 1, Supplementary Figure 9 is redundant and should be removed, Supplementary Figure 11 contains information identical to Figure 1, etc, etc. Finally, I am not sure about the contribution from Supplementary Figure 17. For a biology-heavy manuscript, information in this figure is definitively redundant.

Our response:

We do apologize that there were images perceived to be redundant. We felt that a control image at each time point would be preferred to demonstrate to our readers that the non-transgenic animals were indeed easily imaged and that there was not a strong signal in the appearance of lipid ovoids based on normal aging. However, we agree with Reviewer #3 that some of these images showing similar biological conclusions could be trimmed and revised accordingly, or moved images to supplemental displays.

Reviewer #3 (Remarks to the Author):

4. Supplementary Figure 4 show results measuring the lipid and protein density that correlates with SRS microscopic signal intensity. However, this figure is never cited in the main text, and the results are never explained at all. I believe the data in this figure are important and should be included in Figure 1 along with some histopathology images to establish the correlation between SRS images, biochemical quantification of proteins and lipids, and histopathology characteristics of these lesions in *SOD1G93A*, nerve crush and EAE model.

Our response:

We regret this oversight. We now have referred to **Supplementary Fig. 4** as the evidence that lipid ovoid are indeed lipid-enriched. This question has been partially answered in our response to Point (2) from Reviewer #1 and Point (3) from Reviewer #3. Briefly, we have performed some improved SRS spectrum studies and arranged the main conclusion of spectrum analysis to **Fig. 2b**. To preserve and capture the feature of fixed sciatic nerves from different models, we employed the whole mount immunofluorescence staining (**Methods** of the revised manuscript) on WT, *SOD1G93A*, EAE and sciatic crush animals (**Supplementary Fig. 6** and **Supplementary Fig. 17**). Interestingly, the lipid ovoids visualized by SRS imaging did not resemble immune cells, and did not co-stain with blood-cell specific antibodies (anti-CD45)(**Fig. 2a**). Nor did they frequently contain neuronal components (**Supplementary Fig. 6**). Instead we observed many lipid ovoids co-stained with anti-MBP antibodies and had a chemical composition as measured by SRS highly similar to the known make up of myelin (**Fig. 2a**).

REVIEWERS' COMMENTS:

Reviewer #1 (Remarks to the Author):

In their revised manuscript, the authors have addressed my major concerns. Accordingly, new experimental data (new Figs. 2 and Suppl . 6) have been added as requested, the discussion in the text has been modified, and missing citations have been added.

Unfortunately, the authors did not completely respond to my request No. 3 for providing full experimental details: Because of its relevance for prospective clinical applications, and instead of merely stating on p. 26 (in Methods: In vivo end-point SRS imaging) "... higher laser power was employed for in vivo experiment", please provide the actual pump and Stokes beam powers at the sample for these experiments, too! Furthermore, it is still not clear to me: What were the acquisition times per spectral scan step in the multi-spectral SRS imaging experiments? Please amend.

With the above-mentioned missing experimental details amended, my general opinion about the revised version of this article is positive, and I recommend its publication in Nature Communications.

Reviewer #2 (Remarks to the Author):

My major comments about tuning the stated applications of SRS microscopy as a research tool or tool for ex vivo analysis have been addressed nicely by the authors. They have also acknowledged the issues related the feasibility of the development of a clinical and handheld SRS system in the discussion.

The minocycline treatment experiments are satisfactory with respect to addressing my suggestion that SRS should be evaluated as a means of tracking response to a therapeutic intervention in an ALS model.

The relationship between EMG and SRS for monitoring of ALS-related pathology has also been addressed in the revised discussion.

I would support publication of this manuscript in its current form.

Reviewer #3 (Remarks to the Author):

The authors fully addressed my comments and suggestions from the previous review. Therefore, the revised manuscript is now much improved in its data presentation and organization. I am in full support of the acceptance of the revised manuscript.

Author Response:

Reviewer #1's further concerns regarding the details of microscopic parameters have been addressed by adding descriptive sentences to relevant sections in **Methods**.